# Structural basis of Cdk7 activation by dual T-loop phosphorylation

Robert Düster [1,2], Kanchan Anand[1], Sophie C. Binder [1], Maximilian Schmitz[1], Karl Gatterdam[1], Robert P. Fisher [2] ✉ & Matthias Geyer [1] ✉

Cyclin-dependent kinase 7 (Cdk7) is required in cell-cycle and transcriptional regulation owing to its function as both a CDK-activating kinase (CAK) and part of transcription factor TFIIH. Cdk7 forms active complexes by associating with Cyclin H and Mat1, and is regulated by two phosphorylations in the activation segment (T loop): the canonical activating modification at T170 and another at S164. Here we report the crystal structure of the human Cdk7/Cyclin H/Mat1 complex containing both T-loop phosphorylations. Whereas pT170 coordinates basic residues conserved in other CDKs, pS164 nucleates an arginine network unique to the ternary Cdk7 complex, involving all three subunits. We identify differential dependencies of kinase activity and substrate recognition on the individual phosphorylations. CAK function is unaffected by T-loop phosphorylation, whereas activity towards non-CDK substrates is increased several-fold by T170 phosphorylation. Moreover, dual T-loop phosphorylation stimulates multisite phosphorylation of the RNA polymerase II (RNAPII) carboxy-terminal domain (CTD) and SPT5 carboxy-terminal repeat (CTR) region. In human cells, Cdk7 activation is a two-step process wherein S164 phosphorylation precedes, and may prime, T170 phosphorylation. Thus, dual T-loop phosphorylation can regulate Cdk7 through multiple mechanisms, with pS164 supporting tripartite complex formation and possibly influencing processivity, while pT170 enhances activity towards key transcriptional substrates.

Cyclin-dependent kinase 7 (Cdk7) is a master regulator of cell cycle progression and gene expression, by virtue of its ability to phosphorylate the activation segment, or T loop, of other CDKs, and its essential role in transcription as part of the general transcription initiation factor TFIIH[1]. Cdk7 is found in at least three distinct complexes in cells: a ternary CDK-activating kinase (CAK) complex comprising Cdk7, Cyclin H and Mat1; a quaternary complex of CAK with the TFIIH subunit XPD; and the 10-subunit holo-TFIIH[2–10]. Cdk7 directly influences progression of the cell cycle by phosphorylating and hence activating cell-cycle kinases Cdk1, Cdk2, Cdk4, and Cdk6. As part of TFIIH, Cdk7 is involved in the formation of the pre-initiation complex (PIC) and phosphorylates the carboxy-terminal domain (CTD) of RNA polymerase II (RNAPII) to facilitate promoter escape and RNA 5'-end capping, thereby regulating the expression of mRNA and snRNA[11]. In addition to its direct involvement in transcription via TFIIH, Cdk7 is also an activating kinase for the transcription elongation kinases Cdk9, Cdk12, Cdk13 and probably Cdk11[12–14]. Recently, pharmacologic inhibition of Cdk7 has emerged as a promising option for cancer treatment[15]. Despite longstanding interest in Cdk7, regulation of its kinase activity remains incompletely understood. Early biochemical studies suggested that assembly into TFIIH complexes increased Cdk7 activity towards RNAPII, and that association with Mat1 is sufficient for this stimulatory effect[16–19]. However, Mat1 appears to be stably and

[1]Institute of Structural Biology, University of Bonn, Venusberg-Campus 1, 53127 Bonn, Germany. [2]Department of Oncological Sciences, Icahn School of Medicine at Mount Sinai, New York, NY, USA. ✉e-mail: robert.fisher@mssm.edu; matthias.geyer@uni-bonn.de

constitutively associated with Cdk7 in vivo and is thus unlikely to be a physiologic regulator of kinase activity or substrate specificity[20].

In the case of cell-cycle CDKs, T-loop phosphorylation is strictly and universally required for biological function. The T-loop phosphorylation requirement appears less stringent for Cdk7, as unphosphorylated Cdk7/Cyclin H can form active complexes upon association with Mat1[2,4,21], and phosphorylation-site mutant variants of Cdk7 or its orthologs can support viability in flies and yeast[20,22–24]. These early findings have been validated by recent structural data showing that association with Mat1 shifts the Cdk7 T loop into an active conformation in the absence of phosphorylation[25–27]. Despite being dispensable for basal activation in vitro, T-loop phosphorylation was shown to stimulate Cdk7 activity towards specific protein substrates, including RNAPII and the transcription elongation factor SPT5[12,20,28].

Cdk7 harbors two phosphorylation sites within its T loop. One, T170 (here and throughout, residue numbers refer to the sequence of human Cdk7), corresponds to the canonical site of T-loop phosphorylation typically required for kinase activation, whereas the second phosphorylation site at S164 has not been assigned a clear function. Previous studies indicated redundant or reinforcing roles of dual T-loop phosphorylation in stabilizing the trimeric Cdk7/Cyclin H/Mat1 complex in vitro and in vivo. Preventing Cdk7 T-loop phosphorylation led to loss of complex stability, and to temperature-sensitive lethality in *Drosophila*, with a *cdk7 S164A/T170A* double mutant losing viability at a lower temperature than a *cdk7 T170A* single mutant[20]. However, studies of Cdk7 activity regulation by S164 phosphorylation have been inconsistent, with different groups reporting either kinase activation[20,29] or inactivation[30]. Both S164 and T170 can be phosphorylated in vitro by the Cdk7 targets Cdk1 and Cdk2[4,29,31]. However, no upstream kinase responsible for phosphorylating the Cdk7 T-loop in vivo has been identified, and only a few studies report changes in Cdk7 T-loop phosphorylation in physiologic contexts[13,28,31–34]. Hence, how and when Cdk7 activity is regulated by T-loop phosphorylation remain open questions.

Given the central position of Cdk7 in the cell cycle and RNAPII transcription, elucidation of the pathways that regulate its activity is of crucial importance to understanding both processes. With the emergence of Cdk7 as a potential vulnerability in cancer cells, a deeper understanding of its regulators might yield additional therapeutic opportunities. Despite recent structural advances, however, the molecular basis of Cdk7 regulation remains largely unknown. In this study we determine the structure of the doubly phosphorylated CAK complex, and uncover a previously unrecognized requirement for dual T-loop phosphorylation to stimulate activity towards the transcriptional substrates RNAPII and SPT5. We identify key residues of all three CAK subunits that underlie this effect and define, through biophysical measurements, the contribution of dual T-loop phosphorylation to ternary complex stability. In human colon cancer cells, Cdk7 T-loop phosphorylation is a two-step process in which phospho-S164 appears to be a prerequisite for T170 phosphorylation, revealing unexpected complexity in the regulation of Cdk7 activity in vivo.

## Results

### Crystal structure of the fully active Cdk7/Cyclin H/Mat1 complex

For structure determination, we reconstituted the tripartite human Cdk7/Cyclin H/Mat1 complex comprising Cdk7 (2-346) and Cyclin H (1-323) in complex with a C-terminal fragment of Mat1 (230-309) (Fig. 1a). The position of the $S_{164}P$ motif in Cdk7 is highlighted in a sequence alignment of the T loops of CDKs implicated in transcription (Fig. 1b). A complication in studying Cdk7 T-loop phosphorylation is that the binary complexes produced by co-expression of Cdk7 and Cyclin H in insect cells are phosphorylated at T170 and S164, whereas co-expression of Cdk7, Cyclin H and Mat1 results in ternary complexes that are largely or completely unphosphorylated[20,35]. Therefore, to obtain T-loop phosphorylated ternary complexes, we expressed Cdk7/

Cyclin H or Mat1 separately in insect cells and mixed the two resulting lysates prior to co-purification. These two different purification strategies enabled us to generate and compare wild-type, ternary Cdk7/Cyclin H/Mat1 complexes with or without T-loop phosphorylation (Fig. 1a and Supplementary Fig. 1). The phosphorylation state influences the migration behavior of Cdk7 in sodium dodecyl sulfate-polyacrylamide gel electrophoresis (SDS-PAGE), with the doubly phosphorylated kinase running faster than the unphosphorylated one (Supplementary Fig. 1a–c). The phosphorylation states of the various complex preparations were confirmed by determination of the molecular masses of the intact proteins (Supplementary Fig. 1d–f). When tested for kinase activity towards the RNAPII CTD, the phosphorylated Cdk7/Cyclin H/Mat1$_{230-309}$ was ~3-fold more active than the same complex lacking T-loop phosphorylation, and >5-fold more active than the Cdk7/Cyclin H binary complex (Fig. 1c; compare 7/H/M$_{230-309}$ with 7/H + M$_{230-309}$ and Cdk7/CycH, respectively).

Doubly phosphorylated, fully active CAK complexes were crystallized in the presence of ADP•Mg²⁺ and a chaperoning nanobody, VHH$_{RD7-04}$ (Fig. 1d). The nanobody cDNA was obtained from isolated monocytes after immunization of an alpaca with the ternary complex used for crystallization. Five different nanobodies were identified and characterized by surface plasmon resonance (SPR) spectroscopy for their binding to Cdk7/Cyclin H/Mat1$_{230-309}$, showing affinities between 0.6 and 45 nM and recognition of two different epitopes by different nanobodies (Supplementary Fig. 2a, b). Whereas VHH$_{RD7-01}$ and VHH$_{RD7-05}$ inhibited the kinase activity of Cdk7, the remaining three nanobodies had no effect on the phosphorylation of a CTD substrate (Supplementary Fig. 2c, d). All nanobodies were tested in crystallization trials, but only the Cdk7/Cyclin H/Mat1$_{230-309}$/VHH$_{RD7-04}$ complex grew into crystals that could be optimized to a suitable size using the hanging drop diffusion technique (see Methods). The phases were determined by molecular replacement and the structure refined to a resolution of 2.15 Å with excellent stereochemistry (Supplementary Table 1).

Two Cdk7/Cyclin H/Mat1$_{230-309}$/VHH$_{RD7-04}$ heterotetramers constitute the asymmetrical unit cell of the protein crystal, exhibiting a high overall similarity at an RMSD value of 0.28 Å over 624 C$_\alpha$ atoms. Unexpectedly, residues 1-49 of Cdk7, which form the first three β-strands in the N-lobe of the kinase, are not visible in the electron density map. Instead, the next two canonical β-strands of the kinase, β4 and β5, are in a symmetric crystallographic interface with the opposing Cdk7 subunit, indicating some conformational flexibility within the N-lobe. The nanobody VHH$_{RD7-04}$ interacts through its complementarity determining regions (CDRs) mainly with the C-lobe of Cdk7, with some minor contacts to Mat1 (Supplementary Fig. 2e, f). Overall, the crystallized human CAK complex is similar to the recently reported cryo-EM structure (PDB 6xbz)[25], with an RMSD value of 0.46 Å over 265 C$_\alpha$ atoms. However, while all other Cdk7 structures reported to date contain only one phosphorylation in the T loop (or none), the electron density of the two phosphorylated T-loop residues, pS164 and pT170, is clearly seen (Supplementary Fig. 3a, b), providing structural insights into the regulation of a CDK by dual T-loop phosphorylation.

### Dual phosphorylation keeps the Cdk7 T loop in its place

The two phospho-sites in Cdk7 mediate multiple intra- and inter-subunit interactions that stabilize the conformation of the T loop. The canonical pT170 residue forms salt bridges with R61 of the $^{56}$NRTALREIK αC helix, R136 of the $^{135}$HRD catalytic site motif, and K160 of the $^{155}$DFGLAK substrate interaction motif (Fig. 2a). The formation of this charged cluster is a hallmark of CDK activation[36,37], and in Cdk7 these interactions are evenly distributed with a distance between side chains of 2.5–2.7 Å. pT170 forms an additional hydrogen bond with a water molecule that interacts with the backbone carbonyl of A168. In a pT170-mediated indirect interaction, R61 forms an intermolecular salt bridge with E117 of Cyclin H, which coordinates K64 on the Cdk7 αC

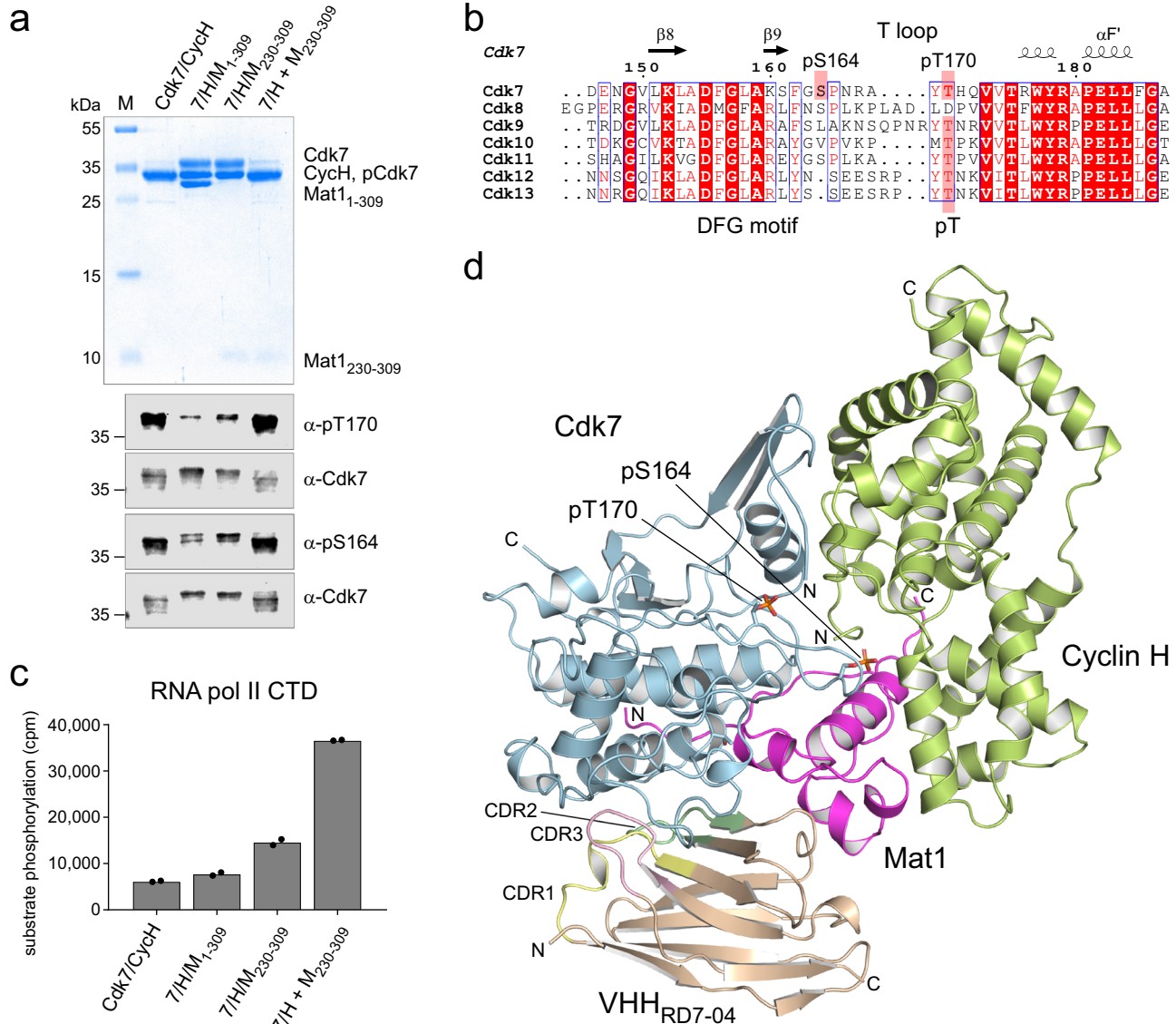

**Fig. 1 | Structure of the doubly phosphorylated Cdk7/Cyclin H/Mat1 complex.**
**a** SDS-PAGE analysis and immunoblotting of Cdk7/Cyclin H/Mat1 complexes. Protein (3 µg) was resolved in a 15% SDS-polyacrylamide gel and stained with Coomassie blue. Note the difference in migration behavior of Cdk7 depending on the phosphorylation status; doubly phosphorylated Cdk7 and Cyclin H migrate at the same size. For immunoblot analysis of the Cdk7 T-loop phosphorylation status, 1 µL protein at 2.6 µM was resolved by SDS-PAGE, transferred to nitrocellulose and probed with phospho-specific antibodies recognizing Cdk7 pT170 or Cdk7 pS164. **b** Alignment of transcriptional CDK T loops. **c**, Radiometric kinase assay probing the preparations for activity towards RNAPII CTD. Each Cdk7 complex (0.1 µM) was incubated with 10 µM GST-CTD[52] in the presence of 1 mM ATP containing 0.35 µCi [$^{32}$P]-γ-ATP for 15 min at 30 °C. Bars represent mean of duplicate measurement. **d**, Crystal structure of the doubly T-loop phosphorylated Cdk7/Cyclin H/Mat1/VHH$_{RD7-04}$ complex at 2.15 Å resolution. Source data are provided as a Source Data file.

helix. Cyclin H residues D116 and E117 that form intermolecular contacts are located at the tip of helix H3 and constitute a di-acidic motif that is highly conserved in cyclins. As a result of this intricate salt-bridge network, the αC helix is pushed toward the active site of the kinase and the DFG motif is in the "in" position, indicative of an active CDK.

The second, Cdk7-specific phospho-site pS164 is buried in the tripartite complex formed with Cyclin H and Mat1. It forms a salt bridge with R165 of the second cyclin box of Cyclin H, which in turn is in a linear arrangement to form another intermolecular salt bridge with D299 of Mat1 (Fig. 2b). The guanidinium ion of R165 of Cyclin H is in a planar stacking relationship with that of Mat1 R295. This stacking is extended to the W132 indole moiety of Cdk7, leading to a parallel alignment of these side chains involving all three subunits (Supplementary Fig. 3c). However, no direct salt-bridge interaction between

Mat1 R295 and pS164 of Cdk7 is seen. Instead, R295 of Mat1 forms a hydrogen bond to the backbone carbonyl of Cdk7 F162 and coordinates four water molecules in proximity to M1 of Cyclin H, S161 of Cdk7, and D299 of Mat1. Remarkably, we found M1 of Cyclin H to be acetylated in the electron density map, explaining the additional 42 Da seen in the mass spectrometry (Supplementary Fig. 1d-f). This residue is fully buried in the ternary complex, with the acetyl moiety interacting with Cdk7 while the methionine side chain points towards Mat1, suggesting that any N-terminal tagging of Cyclin H might impact CAK complex formation. Lastly, the phosphate of pS164 coordinates the side chain amide of N166 in the T loop of Cdk7.

A third residue of Cdk7, R167, centered between pS164 and pT170, coordinates the T loop. It forms an intermolecular salt bridge to D116 of Cyclin H as well as several water-mediated hydrogen bond interactions (Fig. 2a, b). Overall, the high resolution of the crystal structure

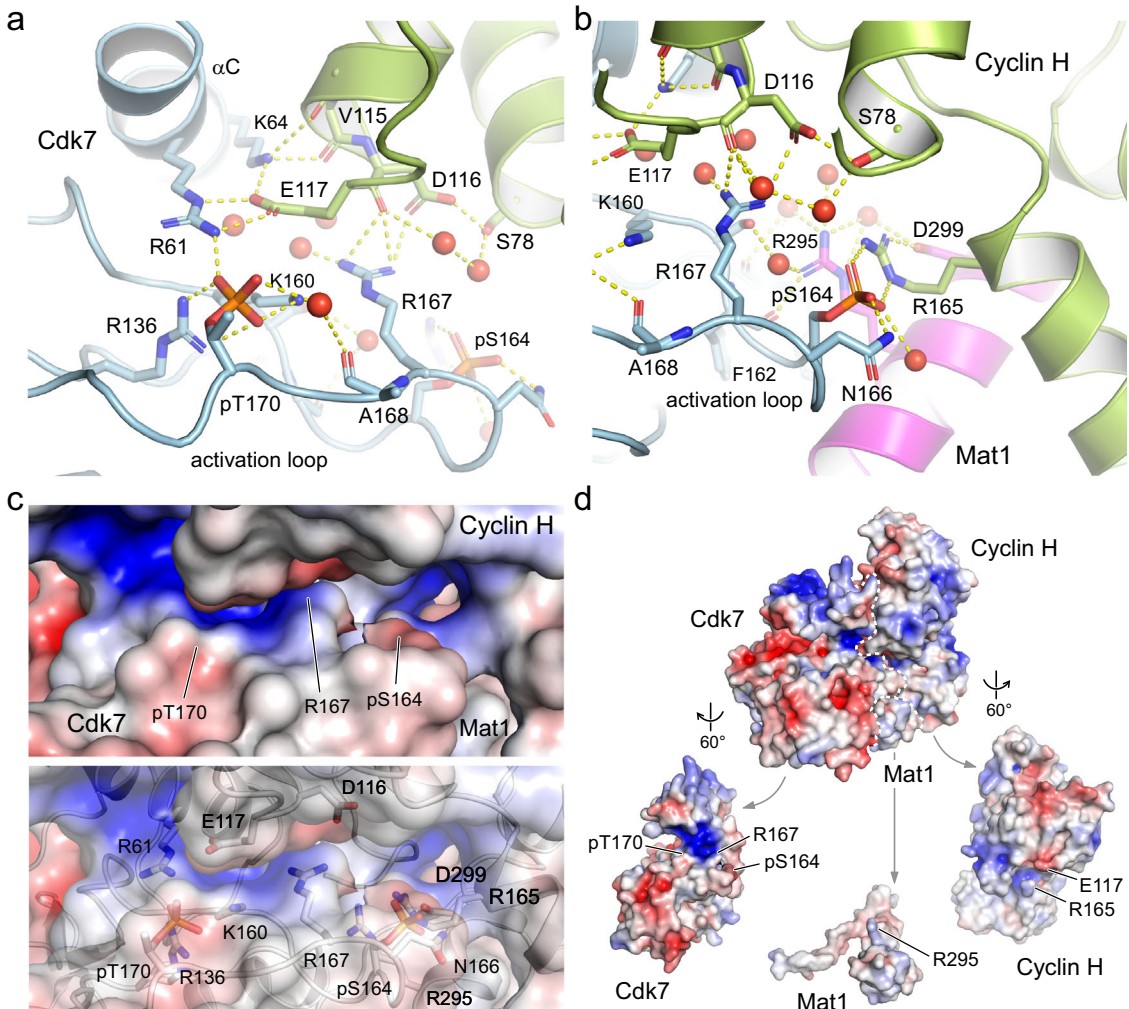

**Fig. 2 | Structural coordination of the phosphorylated T-loop residues in Cdk7.**
**a** Close-up of the interaction network of pT170. Salt bridges are formed to Cdk7 residues R61, R136 and K160, forming the canonical triad of T-loop phosphorylation recognition that is conserved in CDKs. A water-mediated hydrogen bond is formed to the backbone carbonyl of A168. An intermolecular salt bridge between R61 and E117 of Cyclin H extends to K64, coordinating the kinase αC helix. Additional contacts are formed between R167 of Cdk7 and D116 of Cyclin H. **b** Close-up of the interaction network of the non-canonical pS164. An intermolecular salt bridge is formed to R165 on the second cyclin box of Cyclin H, which is continued to D299 of Mat1. The side chain of N166 of Cdk7 is coordinated by pS164. Several water molecules in proximity to the phospho-sites mediate the tripartite interaction. **c** Electrostatic surface display of the T-loop residues in Cdk7. The position of residues pS164, R167 and pT170 is marked. The surface charge and accessibility are visible in the upper panel with the interacting side chains shown in the transparent display of the lower panel. **d** Electrostatics of the ternary Cdk7/Cyclin H/Mat1 complex assembly. The open triptychon display shows charged interactions in the Cdk7/Cyclin H interface while the interactions of Mat1 with both subunits is largely hydrophobic. The electrostatic surface charge is shown from −5 $k_B$T (red) to +5 $k_B$T (blue).

and the many water molecules identified in the interfaces of Cdk7 with Cyclin H and Mat1 provide a detailed picture of the tripartite molecular interaction. The accessibility and electrostatic surface potential of the T-loop residues are shown in Fig. 2c. As these three moieties—pT170, R167 and pS164—shape the surface of the kinase activation loop, we have superimposed an RNAPII CTD heptad repeat on the substrate binding site of Cdk7, based on a Cdk2/Cyclin A/substrate structure[37,38]. Modelling of a $P_3$TSPSYS peptide shows that the two prolines fit well into the characteristic substrate recognition site and that the side chain hydroxyl group of the tyrosine within the $S_5P_6S_7Y_1$ core motif reaches the charged pT170 cluster, allowing for hydrogen bonding (Supplementary Fig. 3d).

The buried surface areas in the ternary CAK complex are evenly distributed, with 2500 Å² for the Cdk7–Cyclin H interface, 2400 Å² for the interaction of Cdk7 with Mat1, and 2400 Å² for Cyclin H and Mat1, counting both molecules. Whereas the interaction between Cdk7 and Cyclin H is highly polar, with the formation of eight salt bridges and more than 20 hydrogen bonds, the interaction of Mat1 with Cdk7 is

mostly hydrophobic, involving no salt bridge and only seven hydrogen bonds. The interaction of Mat1 with Cyclin H is also largely hydrophobic, with two salt bridges and nine hydrogen bonds. The polarity of the interactions can be seen in the electrostatic surface representation, where Mat1 appears largely uncharged and hydrophobic, while complementary positively and negatively charged patches contribute to Cdk7–Cyclin H binding (Fig. 2d).

## S164 phosphorylation is required for Mat1 regulation of Cdk7 activity

To analyze the individual contributions of the two T-loop phosphorylations to Cdk7 activity, we mutated S164 or T170, either individually or in combination, to alanine. Cdk7 variants were expressed and purified in presence of Cyclin H, whereas Mat1$_{230-309}$ was expressed and purified separately as an MBP fusion protein (Fig. 3a). All Cdk7-Cyclin H pairs could be affinity purified to homogeneity with approximately 1:1 stoichiometry, and immunoblot analysis was used to confirm that S164 or T170 were phosphorylated when not mutated. We established

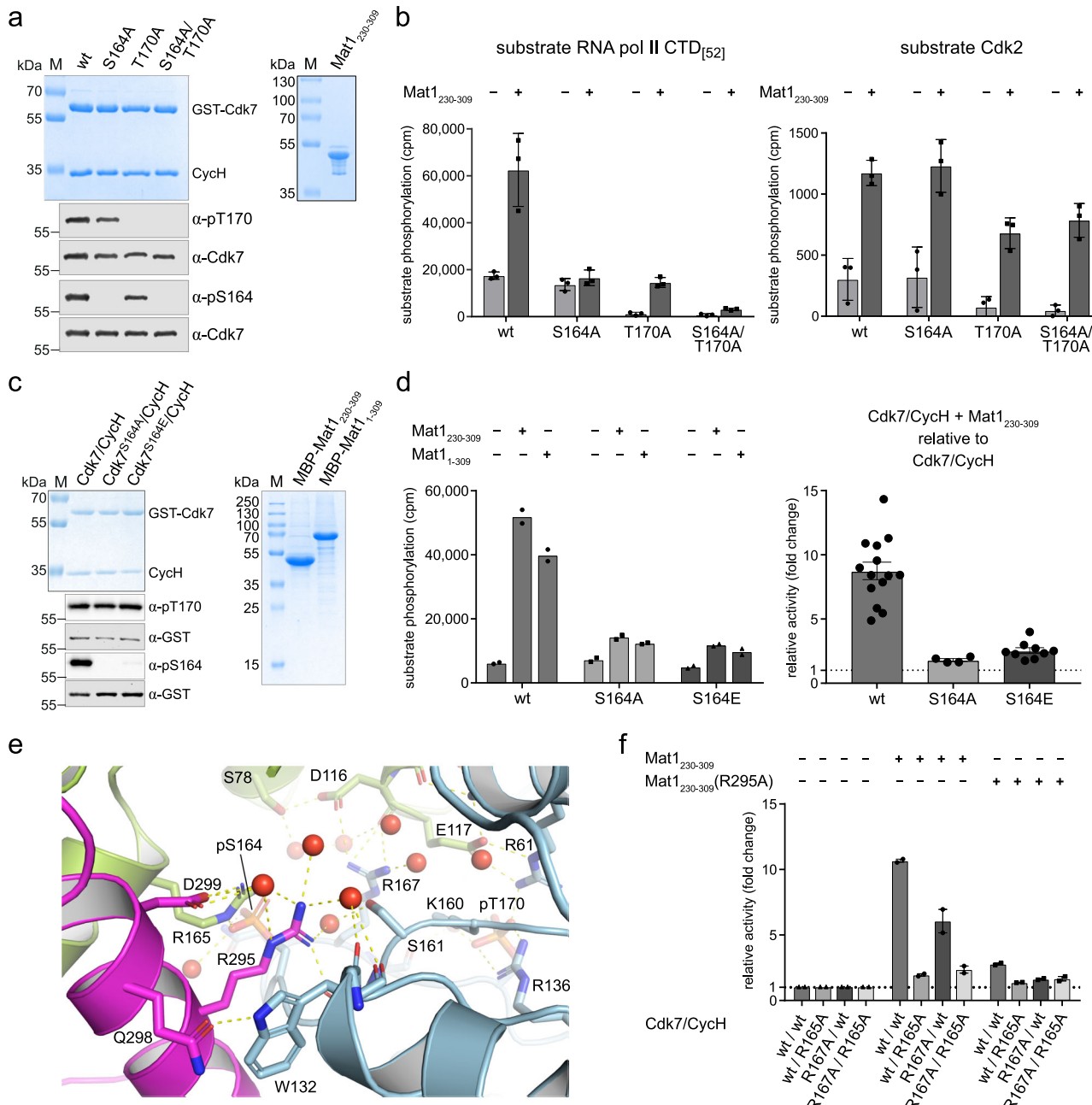

**Fig. 3 | S164 phosphorylation is required for full Cdk7 activity. a** SDS-PAGE and immunoblot analysis of GST-Cdk7/Cyclin H complexes (left panels) and MBP-Mat1$_{230-309}$. The protein (2 µg) was resolved on a 12% SDS gel and stained with Coomassie blue. For analysis of the Cdk7 T-loop phosphorylation status, 1 µL at 2.6 µM was immunoblotted with phospho-specific antibodies recognizing Cdk7 pT170 or Cdk7 pS164. Total Cdk7 was used as loading control. **b** Radiometric kinase assay probing the preparations shown in **a** for activity towards RNAPII CTD and Cdk2 in absence and presence of Mat1. Cdk7/Cyclin H (0.1 µM) was pre-incubated with buffer or 0.4 µM Mat1, and 10 µM GST-CTD$_{[52]}$ or 15 µM GST-Cdk2 for 10 min at room temperature. Reaction was started by addition of 1 mM ATP containing 0.35 µCi [$^{32}$P]-γ-ATP and incubated for 15 min (CTD) or 30 min (Cdk2) at 30 °C. Data represent mean ± SD of triplicate measurements. **c** SDS-PAGE analyses of GST-Cdk7/Cyclin H and MBP-Mat1; 3 µg of each sample was resolved on a 12% SDS-Gel.

Cdk7 T-loop phosphorylation status was analyzed as in **a**. Immunoblotting for GST was used as loading control for Cdk7. **d** Radiometric kinase activity assay. Activity was measured towards GST-CTD$_{[52]}$ as described in **b** (left panel). Data represent mean of duplicate measurements. Fold-change of kinase activity upon MBP-Mat1$_{230-309}$ incubation in relation to the same kinase preparation without Mat1 (right panel). Data represent mean ± SEM; wt, $N = 14$; S164A, $N = 4$; S164E, $N = 9$. **e** Molecular interaction network of R295 of Mat1 with neighboring amino acids. **f** Radiometric kinase activity assay was performed as in **b**. Cdk7, Cyclin H, and Mat1 mutations were as indicated. Data were normalized to the activity of the respective complex without Mat1. Data represent mean ± SEM from two independent experiments, each performed in triplicate. Source data are provided as a Source Data file.

an in-vitro reconstitution assay in which Mat1 was incubated with Cdk7/Cyclin H prior to the start of the kinase reaction (Fig. 3b and Supplementary Fig. 4a, b). This setup allows the contributions of T-loop phosphorylations and Mat1 to kinase activity to be analyzed

separately. Wild-type, doubly phosphorylated Cdk7 was active in the absence of Mat1. The S164 mutation had no effect on Cdk7 activity in the absence of Mat1. In contrast, mutation of T170 to alanine inactivated the kinase. Upon incubation with Mat1$_{230-309}$, the activity of wild-

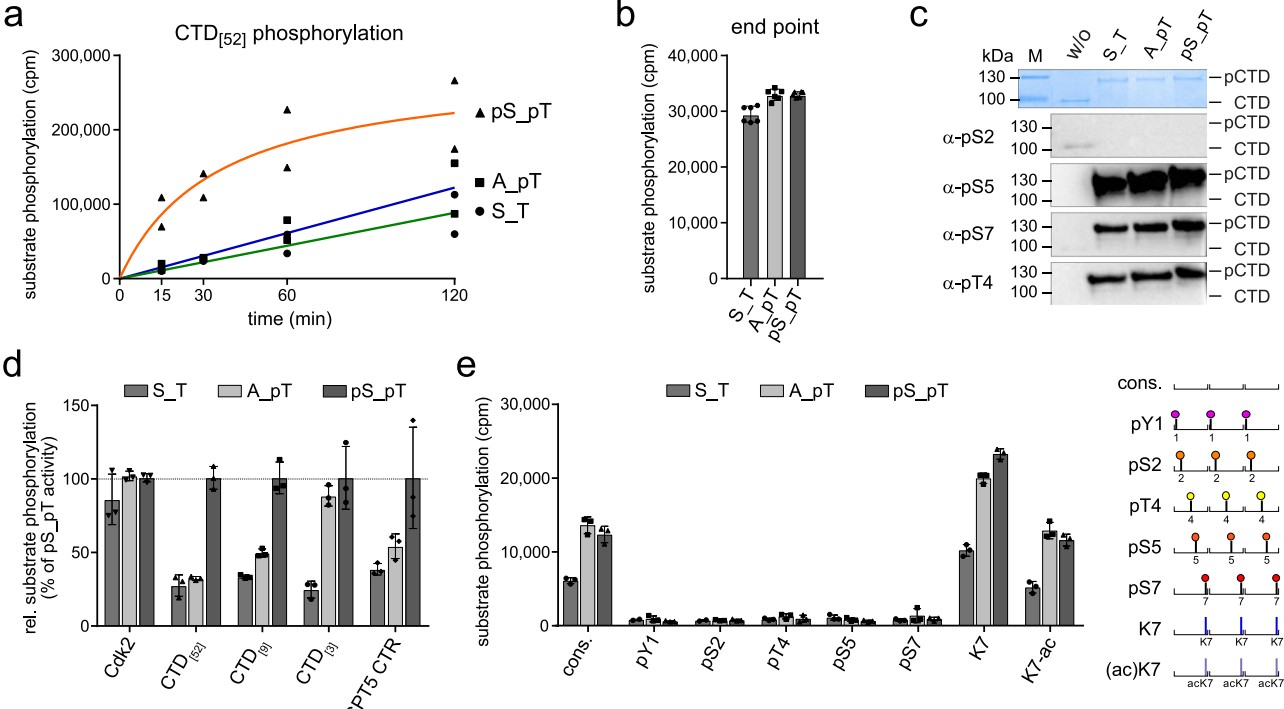

**Fig. 4 | Dual T-loop phosphorylation affects activity but not site specificity of Cdk7 towards the RNAPII CTD. a** Time course measurement of kinase activity of Cdk7/Cyclin H/Mat1 complexes doubly phosphorylated (pS_pT), singly phosphorylated at T170 (A_pT), or non-phosphorylated (S_T), towards GST-CTD$_{[52]}$. Cdk7 complex at 0.1 µM concentration was incubated with 10 µM GST-CTD$_{[52]}$ in the presence of 1 mM ATP containing 0.35 µCi [$^{32}$P]-γ-ATP for indicated times. Curves were obtained by curve fitting using GraphPad prism. **b** Determination of the end-point phosphorylation. The respective Cdk7/Cyclin H/Mat1 complex (1 µM) was incubated with 2.5 µM GST-CTD$_{[52]}$ for 120 min. Data represent mean ± SD of a sextuplicate measurement. **c** Immunoblot analysis of substrate site specificity. Fully phosphorylated GST-CTD$_{[52]}$ (100 ng) was separated by SDS-PAGE, blotted and probed with phospho-specific antibodies recognizing pSer2, pSer5, pSer7, and pThr4 respectively. Migration of hypo-phosphorylated (CTD) and hyper-

phosphorylated (pCTD) forms is indicated. **d** Cdk7/Cyclin H/Mat1 complex (0.1 µM) was incubated with 35 µM GST-Cdk2, 10 µM GST-CTD$_{[52]}$, 50 µM GST-CTD$_{[9]}$, 250 µM CTD peptide (CTD$_{[3]}$), or GST-SPT5$_{748-1087}$ (SPT5 CTR). The activity of pS_pT towards each substrate was set to 100% and the other activities normalized accordingly. Data represent mean ± SEM of triplicate measurements. **e** Cdk7/Cyclin H/Mat1 complex (0.1 µM) was incubated with 250 µM CTD peptides bearing three heptad repeats with either no modification (cons.), consecutive phosphorylations at the indicated site in every CTD repeat (pY1, pS2, pT4, pS5, pS7) or a substitution of the serine at position 7 to lysine. Lysines were either non-modified (K7) or acetylated (K7-ac). Assays were started with 1 mM ATP containing 0.35 µCi [$^{32}$P]-γ-ATP and incubated for 15 min at 30 °C. Data represent mean ± SD of triplicate measurements. Source data are provided as a Source Data file.

type Cdk7 was strongly stimulated. In contrast, Cdk7 S164A activity was not enhanced by Mat1$_{230-309}$. However, whereas Mat1 restored the activity of Cdk7 T170A to roughly the same level as that of Cdk7 S164A and binary, wild-type Cdk7/Cyclin H, the activity of the S164A/T170A mutant was only partially restored, and was below that of non-phosphorylated, ternary complexes containing wild-type Cdk7 (Supplementary Fig. 4c). When the same samples were analyzed for their ability to phosphorylate Cdk2, the requirement for T170 phosphorylation was preserved for binary preparations. However, Mat1 association restored the activity towards Cdk2 in all preparations to a similar degree. Hence, Cdk7 T-loop phosphorylation status is not a major determinant of ternary complex activity towards Cdk2. We obtained similar results when we conducted the experiment with pre-formed ternary complexes that were prepared by co-purification (Supplementary Fig. 4d, e).

To analyze the role of S164 phosphorylation in greater detail, we compared wild-type Cdk7 with Cdk7 S164A or a Cdk7 S164E mutant intended to mimic constitutive phosphorylation (Fig. 3c). As binary complexes with Cyclin H, the three variants did not differ in their T170 phosphorylation status and had similar enzymatic activity, allowing us to discern clearly the contribution of S164 phosphorylation to activity of the ternary complex with Mat1. When incubated with Mat1$_{230-309}$, only wild-type Cdk7 displayed a strong increase in activity, while both the S164A and S164E mutants were only mildly stimulated (Fig. 3d and Supplementary Fig. 4f, g). We repeated the experiment using full-

length Mat1 to exclude artifacts due to N-terminal truncation and obtained similar results (Fig. 3d). Mat1$_{230-309}$ increased the activity of phosphorylated Cdk7 by a factor of 8.8, whereas the S164A or S164E mutants were stimulated only 1.8- or 2.5-fold, respectively (Fig. 3d, right panel).

We hypothesized that the interactions of Cdk7 pS164 with Cdk7 R167 and Cyclin H R165, and its coordination to Mat1 R295, are critical for the observed stimulation of enzymatic activity (Fig. 3e). To test this idea, we mutated Cdk7 R167, Cyclin H R165, and Mat1 R295 to alanine and purified the respective mutant Cdk7/Cyclin H complexes and Mat1$_{230-309}$ R295A. All mutant Cdk7/Cyclin H complexes could be purified to near homogeneity and displayed kinase activity roughly equal to that of wild-type Cdk7/Cyclin H complexes (Supplementary Fig. 4h, i). Upon reconstitution of ternary complexes in vitro, however, either the Cyclin H R165A or Mat1 R295A substitution prevented most of the enhancement of activity by Mat1 addition, recapitulating the effects of the Cdk7 S164A mutation (Fig. 3f). The requirement for Cdk7 R167 was less stringent but still apparent, as its replacement with alanine reduced the stimulation by Mat1 from ~10- to ~5-fold. The requirement for a coordinating network provides a plausible explanation of why the Cdk7 S164E mutation was unable to mimic S164 phosphorylation; the single negative charge introduced by a glutamate would not suffice to mediate the multivalent interactions needed for full activation. Taken together, these data indicate that phosphorylation of Cdk7 S164 is required to form an electrostatic interaction

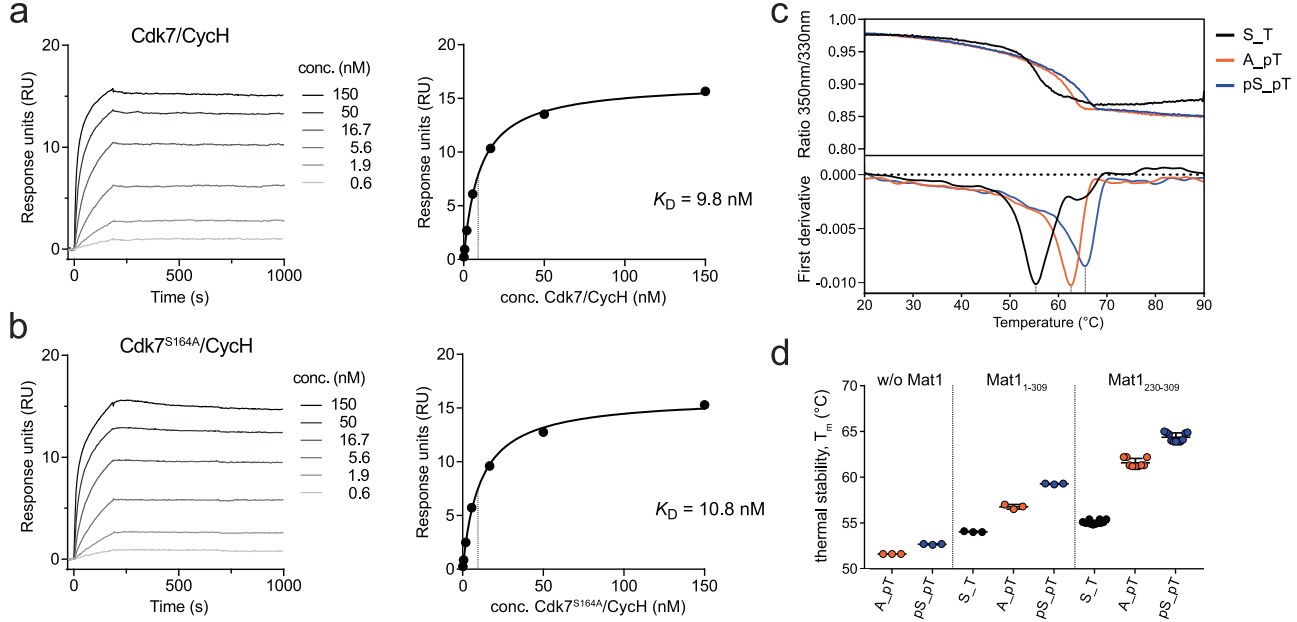

**Fig. 5 | Mat1 binds tightly to Cdk7/Cyclin H and synergizes with Cdk7 T-loop phosphorylation to stabilize the tripartite complex. a, b** Multi-cycle kinetics of SPR measurements. For analysis, Cdk7/Cyclin H complex was used as analyte in a serial 1:3 dilution ranging from 150 nM to 0.2 nM (left panels). Dissociation constants were calculated by determination of the steady-state affinity (right panels). **c** Thermal stability of Cdk7 complexes was determined at a protein concentration of 2.5 μM in storage buffer (20 mM Hepes pH 7.6, 150 mM NaCl, 1 mM TCEP) by monitoring intrinsic fluorescence at 350 and 330 nm with a nanoDSF device (NanoTemper). The chromatogram shows the melting curves of the ternary complexes indicated. **d** Dotplot representation of the melting point ($T_m$) of binary and ternary Cdk7 preparations. Stability was monitored in triplicate (samples w/o Mat1 and with Mat1$_{1-309}$) or comprising twelve (S_T), nine (A_pT) and eleven (pS_pT) replicates (samples with Mat1$_{230-309}$). Data represent mean ± SD. Source data are provided as a Source Data file.

network with the other CAK subunits, which is in turn needed for Mat1-dependent kinase stimulation (Fig. 3f).

## T-loop phosphorylation regulates Cdk7 activity towards different substrates

We further characterized the activity profiles of ternary Cdk7/Cyclin H/Mat1$_{230-309}$ complexes in which Cdk7 is either non-phosphorylated (S_T), singly phosphorylated at T170 (A_pT), or doubly phosphorylated at S164 and T170 (pS_pT). The singly phosphorylated variant pS_A was not tested because the S164 phosphorylation level in the T170A mutant was <20% that of wild-type Cdk7, as determined by western blot analysis. As shown above, doubly phosphorylated Cdk7 possesses higher activity towards the RNAPII CTD, whereas phosphorylation at Cdk7 T170 alone has only minimal effects on activity (Fig. 4a). The $K_{M(ATP)}$ was similar for all three kinase preparations (S_T, 51.7 μM; A_pT, 40.3 μM; pS_pT, 75.5 μM) and could not explain the differences in activity observed at 1 mM ATP (Supplementary Fig. 5a). Likewise, the $K_{M(CTD)}$ was not affected by the T-loop phosphorylation status of Cdk7 (S_T, 24.3 μM; A_pT, 31.2 μM; pS_pT, 31.9 μM) (Supplementary Fig. 5b, c). We did not observe differences in total CTD phosphorylation at saturation, suggesting that dual T-loop phosphorylation increases the catalytic rate, as previously reported[20], but not the total number of repeats that become phosphorylated (Fig. 4b). Analysis with phosphosite-specific antibodies revealed no differences in the amount of CTD Ser5 and Ser7 phosphorylation at saturation (Fig. 4c). Moreover, addition of Mat1 stimulated activity of binary Cdk7/Cyclin H complexes towards both Ser5 and Ser7 but did not boost the pSer2 signal above the limit of detection (Supplementary Fig. 5d). This suggests that neither T-loop phosphorylation nor Mat1 binding shifts Cdk7's phosphorylation site preferences within CTD repeats. Interestingly, while Ser5 and Ser7 are known targets of phosphorylation by Cdk7, we also detected phosphorylation of Thr4, which has not been previously reported (Fig. 4c and Supplementary Fig. 5e). In a control reaction, incubation with Cdk9/Cyclin T1 produced a phosphorylation pattern on the CTD similar to those previously described: strong signals with anti-pSer5 and -pSer7, and weak or undetectable signals with anti-pSer2 and -pThr4[39–41].

As observed above (Fig. 3b), the CAK activity of Cdk7 is largely insensitive to the T-loop phosphorylation status (Fig. 4d and Supplementary Fig. 5f–i). The striking difference in T-loop phosphorylation dependency between CTD and Cdk2 substrates led us to explore other Cdk7 substrates. The strong enhancement of activity towards the full-length RNAPII CTD by pS164 was preserved on a shorter GST-CTD construct containing nine consensus repeats (GST-CTD$_{[9]}$). However, with a peptide substrate consisting of three consensus repeats, T170 phosphorylation resulted in increased activity that could not be further enhanced by additional S164 phosphorylation (Fig. 4d and Supplementary Fig. 5f, g). Interestingly, a substrate containing the C-terminal repeat regions (CTRs) 1 and 2 of SPT5−repetitive motifs superficially similar to the RNAPII CTD, with a total of 29 sites matching the partial CDK phosphorylation consensus (S/T-P)−resembled full-length or nine-repeat CTD substrates in its response to pS164. This is consistent with previous data identifying the SPT5 CTRs as Mat1-dependent Cdk7 substrates in vitro[12] and suggests that S164 phosphorylation selectively stimulates Cdk7 activity towards multi-site substrates.

We next tested if CTD phosphorylation by Cdk7 is primed by other CTD phosphorylations present in the same repeat. We analyzed a set of peptides containing three heptad repeats designed to mimic known CTD modifications or amino-acid substitutions, including phosphorylations, acetylations or the common Ser7-to-Lys substitution[42]. Any uniform, pre-existing phosphorylation prevented Cdk7 from placing additional phosphates on the CTD peptides, whereas phosphorylation of a single repeat reduced phosphorylation of adjacent repeats (Fig. 4e and Supplementary Fig. 5j). This behavior contrasts with that of Cdk9, Cdk12, Cdk13 and DYRK1a, which are capable of phosphorylating (or even prefer) certain pre-phosphorylated CTD species, and which act downstream of Cdk7 in

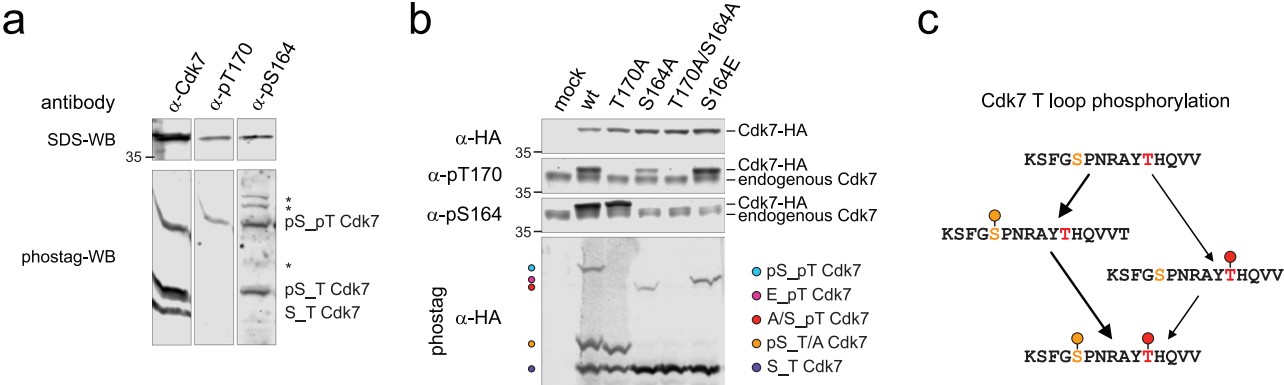

**Fig. 6 | Phospho-S164 enhances T170 phosphorylation of Cdk7 in human cells.**
**a** Phos-tag-SDS-PAGE and conventional SDS-PAGE immunoblot analysis of HCT116 whole-cell lysate. Membranes were probed with α-Cdk7 antibody to visualize total Cdk7 and in parallel with phospho-specific antibodies recognizing Cdk7 pT170 or Cdk7 pS164. Unspecific bands are marked by asterisks. **b** Conventional immunoblot and phos-tag immunoblot analysis of HCT116 cell lysates transfected with Cdk7-HA T-loop mutants. Blots were probed with α-HA antibody to visualize transfected Cdk7-HA and in parallel with phospho-specific antibodies recognizing Cdk7 pT170 or Cdk7 pS164. **c** Theoretical Cdk7 T loop phospho-isoforms, indicating possible pathways of sequential phosphorylation of S164 and T170.

transcription[40,41,43–45]. It is consistent, however, with the ability of fission yeast Mcs6, the ortholog of Cdk7, to prime CTD substrates for subsequent phosphorylation by Cdk9, but not vice versa[46]. Substitution of Ser7 by lysine enhanced Cdk7 activity approximately twofold, but this effect was abolished when the lysines were acetylated (Fig. 4e).

### T-loop phosphorylation increases ternary complex stability but not Mat1 binding affinity

To determine if Cdk7 S164 phosphorylation influences the formation of ternary complexes with Mat1, we established a surface plasmon resonance (SPR) spectroscopy assay. We covalently coupled an anti-maltose binding protein (MBP) antibody to a CM5 chip, which allowed us to immobilize MBP-Mat1$_{230-309}$ (Supplementary Fig. 6a). We then applied either Cdk7/Cyclin H, Cdk7(S164A)/Cyclin H or Cdk7(S164E)/Cyclin H as analytes. The binding of Mat1$_{230-309}$ to Cdk7/Cyclin H complexes was analyzed in multi-cycle kinetics experiments at concentrations ranging from 0.2 to 150 nM (Fig. 5a, b). Cdk7/Cyclin H association with Mat1$_{230-309}$ appeared to be slow, consistent with the lack of electrostatic forces driving the interaction. However, once complexes formed, we observed little dissociation, indicating high stability consistent with the large interaction surfaces involving all three subunits. We determined a steady-state affinity ($K_d$) of 9.8 nM for the complex formation between wild-type Cdk7/Cyclin H and Mat1$_{230-309}$ (Fig. 5a). Binding kinetics and affinity of Cdk7(S164A)/Cyclin H and Cdk7(S164E)/Cyclin H complexes were indistinguishable from those of the wild-type enzyme, indicating that Cdk7 S164 phosphorylation does not contribute to ternary complex formation directly (Fig. 5a, b and Supplementary Fig. 6b).

Previous studies reported a strong impact of Mat1 binding and T-loop phosphorylation on complex stability[4,20]. We therefore compared the thermal stability of binary and ternary Cdk7 preparations with differing T-loop phosphorylation status over a temperature gradient ranging from 20–90 °C (Fig. 5c). Stability of full-length ternary complexes increased by 2.8 °C upon T170 mono-phosphorylation and by another 2.5 °C upon dual S164/T170 phosphorylation (Fig. 5d). A contribution of S164 phosphorylation to stability was consistent among the complexes we tested; even in the absence of Mat1, the binary complex with dual S164/T170 phosphorylation was slightly more stable (by 1.0 °C) than one bearing only the pT170 mark. The stabilizing effects were even more pronounced for complexes with truncated Mat1$_{230-309}$, with an increase in thermal stability of 9.4 °C between the unphosphorylated and fully phosphorylated forms.

### Phospho-S164 promotes T170 phosphorylation in HCT116 cells

The two T-loop phosphorylation sites allow for four distinct Cdk7 phospho-isoforms, with potentially two alternative paths from the non-phosphorylated to the fully phosphorylated state. To resolve the phospho-isoforms, we used Phos-tag-SDS-PAGE followed by western blot analysis. The Phos-tag reagent binds to phosphorylated proteins, altering their migration and allowing for the separation of phospho-isoforms in reducing SDS-PAGE (Supplementary Fig. 7a). Conventional western blot analysis confirmed the presence of both pS164 and pT170 on Cdk7 in HCT116 whole cell lysate (Fig. 6a). Separation of the lysate by Phos-tag-SDS-PAGE, followed by western blotting, revealed three Cdk7 phospho-isoforms present within the lysate. Probing with a pT170-specific antibody showed that only the slowest-migrating form carries the T170 phosphorylation. In contrast, pS164 was present in both the slower-migrating and intermediate Cdk7 phospho-isoforms. The fastest-migrating form did not contain either phosphorylation; taken together with previous results suggesting that nearly all cellular Cdk7 is in ternary or higher-order complexes[2,4,14,20], this might indicate a role for Mat1 in maintaining complex stability in vivo in the absence of T-loop phosphorylation. Moreover, the absence of a Cdk7 isoform monophosphorylated on T170 suggested that T170 phosphorylation might depend on the prior phosphorylation of S164.

If Cdk7 S164 phosphorylation promotes T170 phosphorylation, a S164A mutant would be predicted to have reduced T170 phosphorylation. To test this hypothesis, we transfected HCT116 cells with plasmids expressing different, HA-tagged Cdk7 variants (wild-type or T-loop mutants), and analyzed their T-loop phosphorylation status (Fig. 6b). Wild-type Cdk7-HA was phosphorylated at both sites, S164 and T170. Mutation of T170 to alanine did not affect pS164 levels. In contrast, mutation of S164 to alanine strongly reduced phosphorylation at T170. Replacement of S164 with glutamate mostly restored the level of T170 phosphorylation as found in wild type Cdk7-HA (Fig. 6b and Supplementary Fig. 7b, c). When analyzed by Phos-tag-SDS-PAGE and western blotting, phospho-isoform distribution for wild type Cdk7-HA recapitulated the results for endogenous Cdk7, with mono-phosphorylation of S164 but not T170 in addition to a doubly phosphorylated isoform (Fig. 6b and Supplementary Fig. 7d, e). Together, these data suggest a stepwise model of Cdk7 T-loop phosphorylation in HCT116 cells, in which S164 is phosphorylated first and either primes T170 phosphorylation or stabilizes pT170 by promoting a conformation more resistant to attack by cellular phosphatases (Fig. 6c).

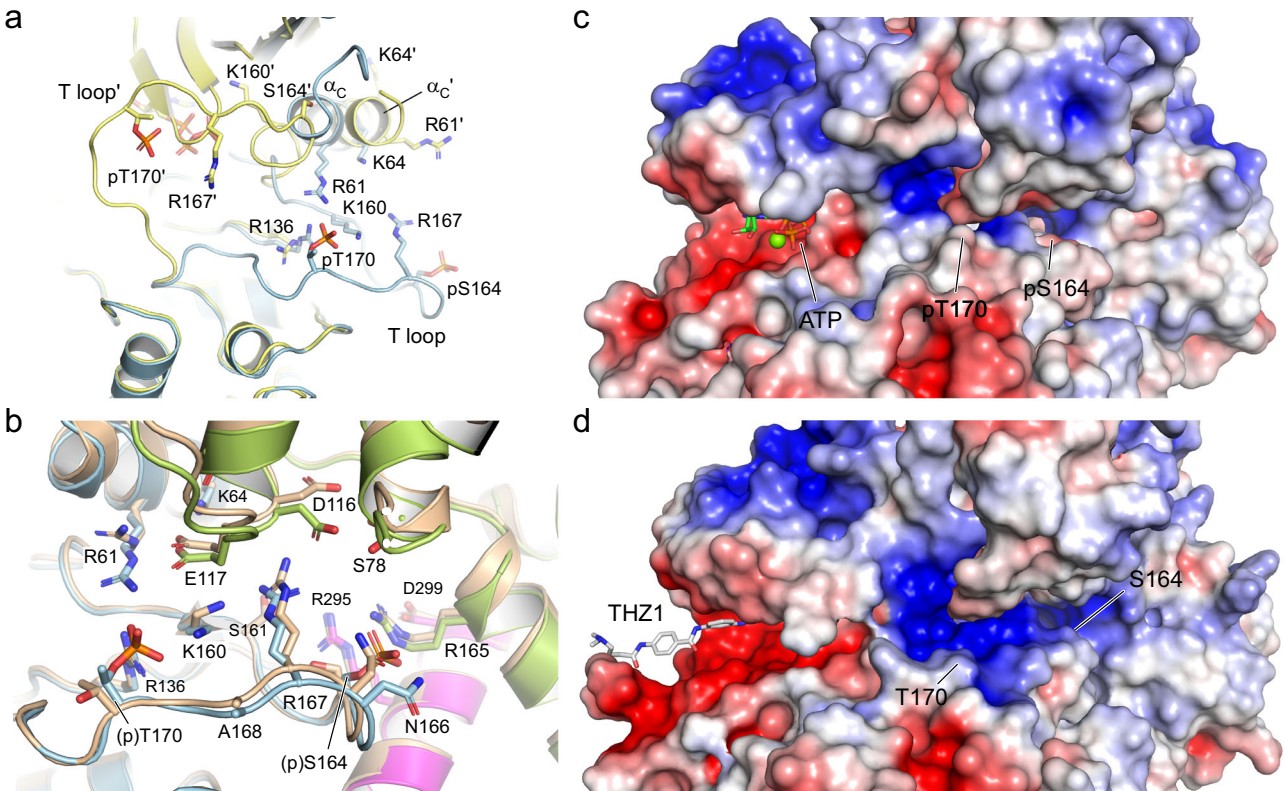

**Fig. 7 | Comparison of T-loop phosphorylated and unphosphorylated Cdk7 structures. a** Superimposition of the crystal structure of T170 phosphorylated monomeric Cdk7 (PDB 1ua2; yellow)[29] with the doubly T-loop phosphorylated Cdk7 structure determined here (8pyr, blue). Despite T170 phosphorylation, Cdk7 adopts a completely different conformation of the T loop in the absence of Cyclin H and Mat1. **b** Overlay of the unphosphorylated CAK structure (8orm, sand)[47] with the doubly phosphorylated complex structure (8pyr). Only subtle differences exist in the positioning of the side chains, while the conformation of the T loop is largely unchanged. **c** Display of the electrostatic surface potential for the doubly phosphorylated CAK structure (8pyr). The phosphate groups of pS164 and pT170, each carrying two negative charges, oppose the basic residues and neutralize the surface charge. ATP•Mg²⁺ and the N-terminal residues 10-54 of Cdk7 were modelled into the CAK complex as described in Supplementary Fig. 3d. **d** The same view on the unphosphorylated CAK structure bound to the inhibitor THZ1 (8orm)[47] reveals an extended basic surface patch. The electrostatic surface charge is shown from $-5\,k_{B}T$ (red) to $+5\,k_{B}T$ (blue).

## Discussion

Here we determined, at 2.15 Å resolution, the crystal structure of the human Cdk7/Cyclin H/Mat1 complex with the kinase doubly phosphorylated at S164 and T170 in the T loop. Mat1 binding seals the cleft between Cdk7 and Cyclin H at the site of the kinase activation segment, and extends the common interface formed by CDK–cyclin binding through binary interactions with the kinase C-lobe and the second cyclin box of Cyclin H. These interactions are mostly hydrophobic in nature and evenly distributed among all three subunits of the CAK complex, indicative of a tight ternary assembly with high thermal stability. The determination of Cdk7 structures in unphosphorylated, singly T170 phosphorylated, and doubly T-loop phosphorylated states allows the comparison of these three different forms. In the absence of Cyclin H and Mat1, the T loop of Cdk7 was found to adopt a completely different conformation wherein the salt bridges characteristic of CDKs (to R61, R136, and K160) were not formed despite T170 phosphorylation (Fig. 7a)[29]. The recent high-resolution cryo-EM structures of human CAK in complex with various Cdk7-specific inhibitors allow comparison with the structure of the kinase with an unphosphorylated T loop[47,48]. Superimposition of the THZ1-bound Cdk7/Cyclin H/Mat1 complex structure (PDB ID 8orm) with the doubly phosphorylated CAK structure determined here reveals only subtle differences in the positioning of the side chains while the conformation of the T loop is largely unchanged, suggesting that Cyclin H and Mat1 are sufficient to position the Cdk7 T loop within the ternary complex and support basal kinase activity (Fig. 7b). However, a comparison of the electrostatic

surface potentials shows significant differences between these two differently active forms. Whereas the negatively charged phosphate groups of pS164 and pT170 oppose the basic residues of Cdk7, Cyclin H, and Mat1, neutralizing the overall surface charge, the unphosphorylated CAK structure exhibits a largely basic surface patch at this position (Fig. 7c, d). Notably, this site adjacent to the kinase active center has been shown to be involved in substrate recognition in other CDKs, through salt-bridge formation of the S/T-P-x-R recognition motif with the canonical T loop phospho-site[37,38,49].

Establishment of the two T-loop networks is required for maximal Cdk7 activity towards transcriptional substrates, indicating regulatory functions for both pS164 and pT170. The doubly phosphorylated ternary complex had higher activity towards the RNAPII CTD and the SPT5 CTRs, which both contain repetitive iterations of S/T-P motifs. This substrate-specific effect has been reported previously but ascribed only to the canonical T-loop phosphorylation at T170[12,20]. We found that phosphorylation at both S164 and T170 is required for enhanced activity towards longer (and likely more physiologically relevant) substrates; for the RNAPII CTD, the threshold above which pS164 exerts a stimulatory effect is between three and eight repeats (each with as many as three potential sites of phosphorylation). An interesting possibility is that, by neutralizing the positive charge on the surface adjacent to the active site, dual T-loop phosphorylation weakens interactions with already phosphorylated CTD repeats and enhances kinase processivity by promoting product release. Besides its direct effect on enzyme activity, pS164 promoted or stabilized T170

phosphorylation in human cells, suggesting another mechanism by which dual T-loop phosphorylation positively regulates Cdk7 activity.

S164 and other network components are conserved among higher eukaryotes, but missing or imperfectly conserved in *S. cerevisiae*, *S. pombe* and *Chaetomium thermophilum*. In the budding yeast Cdk7 ortholog, Kin28, the residue equivalent to S164 is replaced by alanine. *S. pombe* Mcs6, on the other hand, contains two potential phosphorylation sites in similar positions within the T loop, with the non-canonical site residing within a partial CDK consensus motif (TP). In the recently determined structure of the Cdk7/Cyclin H/Mat1 complex of *Chaetomium thermophilum*, S164 is replaced by an aspartate (D247) that interacts with R190 of the cyclin partner (which aligns with R165 of human Cyclin H), but the dependency of kinase activity on this residue was not analyzed[26]. There have been sporadic reports of non-canonical T-loop phosphorylations influencing activity of other CDKs. In the case of human Cdk9, an inhibitory function was ascribed to phosphorylation at S175, eleven amino-acid residues away from the activating T186 residue[50]. Interestingly, in addition to a canonical activating threonine (T591), the Cdk11 T loop contains a serine (S585) in a perfect consensus site for CDK recognition, similar to the site harboring Cdk7 S164 (Fig. 1b). Phosphorylation of Cdk11 S585 is described in public databases, but has not been functionally characterized. A recent study identified a binding partner of Cdk11/Cyclin L, SAP30BP, which stabilized and activated the complex, possibly hinting at the regulation of Cdk11 activity through ternary complex formation, similar to that described here for Cdk7[51].

Having identified a molecular network within the ternary Cdk7 complex required for full kinase activation, we can envision how this arrangement might promote multi-site CTD phosphorylation. Modeling of a CTD substrate peptide onto the Cdk7/Cyclin H/Mat1 complex with $S_5$ occupying the kinase active site, based on a superimposition with the structure of Cdk2/Cyclin A bound to a PK**TP**KK**K**AKKL substrate[38], shows $Y_1$ of the next repeat aligned perfectly to form a hydrogen bond with pT170 (Supplementary Fig. 3d). The pS164 cluster, however, is buried in the tripartite complex assembly and might only contribute indirectly to the CTD substrate interaction by restructuring the surface layer in the inter-subunit interface. A multiply phosphorylated CTD substrate, resulting from processive phosphorylation by the kinase, is highly negatively charged and would require a positively charged counterpart for strong association. Moving further along the docked CTD in a C-terminal direction, a highly basic patch at the backside of Cyclin H, formed by residues R197, R223, K253, R256 and K260, could make electrostatic interactions with the phosphorylated CTD (Supplementary Fig. 8). As this site is over 50 Å away from the kinase active site, it might require four or more CTD repeats to bridge the distance to this surface patch. In support of this hypothesis, in the structure of the Mediator-bound PIC, Cdk7 is positioned between the Mediator and TFIIH subunits and cannot freely approach CTD repeats for phosphorylation; the CTD repeats are processively fed through the active site of the kinase, most likely resulting in an N- to C-terminal direction of phosphorylation[52]. Future experiments will be needed to determine whether the disposition of these basic residues is influenced by the pS164-centered electrostatic network, possibly explaining why the pS164 effect on processivity is not retained with CTD peptides harboring only three repeats.

Cdk7 was unable to phosphorylate CTD repeats that already contained a phosphorylation at any site. This is in contrast to many other transcriptional kinases that tolerate or even prefer pre-phosphorylation within the same or adjacent repeats[40,41,43-46,53]. This limitation comports, however, with Cdk7's role as the first kinase to phosphorylate the CTD during transcription initiation, and might prevent Cdk7 from acting ectopically during transcription elongation.

The observation that Cdk2 phosphorylation is insensitive to the Cdk7 T-loop phosphorylation status is in line with earlier studies and supports the conclusion that positioning of T160 of Cdk2 within the Cdk7 active site is not primarily T-loop sequence-dependent, but requires recognition of distal structural elements[12,31,54]. Moreover, it is in agreement with findings that CAK activity in extracts is constant during the cell cycle of continuously dividing cells[21,55]. In an apparent exception to this general rule, Cdk7 T170 phosphorylation, which is down-regulated in quiescent cells, increases in chromatin-associated Cdk7 upon cell-cycle re-entry, and can stimulate activation of Cdk4 by Cdk7 in vitro[28].

Early studies reported that T170 phosphorylation requires translocation of Cdk7 to the nucleus and that both T-loop sites are phosphorylated in *Xenopus laevis* extracts[56,57]. Phosphorylation of T170 (T176 in *X. laevis*) was shown to be the major determinant for Cyclin H binding in the absence of Mat1, which is in line with studies from human cells[4]. Phosphorylation of the serine alone enhanced Cdk7–Cyclin H interaction but was not sufficient to promote stable complex formation[4,57]. This observation can be explained by the interaction of pS164 with R165 of Cyclin H and is reflected in the contribution of S164 phosphorylation to complex stability. In contrast to Cyclin H binding, Mat1 association with Cdk7/Cyclin H is not affected by the S164 phosphorylation status, and Mat1 associates with non-phosphorylated Cdk7/Cyclin H or even S164A/T170A complexes (Fig. 5a, b and Supplementary Fig. 4d). This phosphorylation-independent binding reflects the large, mainly hydrophobic interface of Mat1 with Cdk7/Cyclin H, which contrasts with the many polar interactions between Cdk7 and Cyclin H. Cdk7 T-loop phosphorylation may nonetheless contribute to ternary complex formation by stabilizing a transient, binary Cdk7/Cyclin H complex that might provide a better scaffold for Mat1 association. Accordingly, *Drosophila cdk7* $^{S164A/T170A}$ mutants show severely impaired ternary CAK complex stability and temperature-sensitive lethality[20].

The pathways responsible for Cdk7 phosphorylation in vivo have not yet been characterized, and candidate Cdk7-activating kinases such as CDKs or Protein Kinase C (PKC) isoforms have not been confirmed. The sequence adjacent to Cdk7 S164 (SPNR) matches the perfect CDK/MAPK consensus sequence (S/T-P-x-R), suggesting regulation by mitogenic stimulation or cell cycle-intrinsic signals. Accordingly, Cdk1 and Cdk2 have been shown to phosphorylate both S164 and T170 (which does not match even the partial CDK consensus, S/T-P) in vitro[4,29,31,57]. Studies that describe changes in Cdk7 T-loop phosphorylation upon different perturbations often report concomitant deregulation of the cell cycle[32,34,58]. However, whether the observed changes in Cdk7 T-loop phosphorylation occurred as a consequence or cause of cell-cycle deregulation was not clearly determined. Another interesting possibility is that Cdk7 T-loop phosphorylation is responsive to growth factor signaling; in a recent report, cells with activating mutations in the phosphatidylinositol-4,5-bisphosphate 3-kinase (PIK3CA), which result in hyperactive growth signaling, had increased sensitivity to Cdk7 inhibition[33,59].

Based on our results, Cdk7 T-loop phosphorylation appears to be a sequential, two-step process in which Cdk7 S164 phosphorylation is established first, followed by T170 phosphorylation. In whole-cell extracts prepared under conditions designed to solubilize chromatin-associated proteins, doubly phosphorylated Cdk7 represents only a minor fraction of total Cdk7, with the majority being unphosphorylated or mono-phosphorylated at Ser164 (Fig. 6b). As we show here, dual Cdk7 T-loop phosphorylation is required to boost Cdk7 activity towards multi-site phosphorylation substrates involved in transcription. Disruption of cell cycle-regulated transcriptional programs is one mechanism by which covalent Cdk7 inhibitors have been suggested to act selectively on cancer cells[60]. Combined with evidence that cell-cycle regulation of Cdk7 occurs predominantly in the chromatin-bound subpopulation of Cdk7[12,20,28], this raises the interesting possibility that Cdk7 T-loop phosphorylation specifically regulates the transcription of cell-cycle genes.

## Methods

### Generation of multi-gene-expression vectors

All expression plasmids were generated by restriction enzyme-based cloning. Restriction enzyme cleavage sites were introduced by PCR. All plasmid inserts were confirmed by Sanger sequencing. Plasmids containing multiple genes for co-expression in *Spodoptera frugiperda* 9 (*Sf*9) insect cells were generated using the MultiBac system (Geneva Biotech) using pACE-Bac1 vectors modified in house with N-terminal MBP or GST affinity tags followed by a TEV protease cleavage site. The vectors were fused with pIDK and pIDC donor vectors by Cre recombination. Successful Cre-recombination was validated by antibiotic selection and the stoichiometry of the fusion vector confirmed by restriction enzyme digestion.

### Site directed mutagenesis

Point mutations were introduced into plasmids by primer-directed mutagenesis in a PCR reaction. The non-mutated parental plasmid was digested by the methylation-sensitive restriction endonuclease *Dpn*I. Point mutations were confirmed by sequencing. Primers for site-directed mutagenesis consisted of approximately 45 nucleotides carrying the mutation in the center. Parental plasmids were digested by adding 5 μL 10x cutsmart buffer (NEB), and 1 μL *Dpn*I (NEB) directly to the PCR reaction mixture after PCR and continued incubation for 1 h at 37 °C. After restriction digestion, 5 to 10 μL of the reaction mixture was used to transform NEB β10 cells.

### Protein expression

Recombinant proteins were expressed in baculovirus infected *Sf*9 cells (Cdk7, Cyclin H, Mat1) or in *Escherichia coli* BL21 (DE3) pLysS bacterial cells (GST-CTD$_{[52]}$, GST-CTD$_{[9]}$, GST-SPT5, GST-Cdk2). For expression in *Sf*9 cells, cells at a density of $1.5 \times 10^6$ cells/ml were infected by adding 2% (v/v) of baculovirus V2 preparation. Cells were harvested after 72 h by centrifugation at $1000 \times g$ (JLA8.1 rotor, Beckman-Coulter), washed with PBS, snap-frozen in liquid nitrogen, and stored at −20 °C or −80 °C. For expression of proteins in bacteria, *E. coli* cells were grown in LB medium containing appropriate antibiotics at 37 °C (pre-culture). The next day, optical density at 600 nm (OD$_{600}$) was determined and the pre-culture was diluted into larger volumes of LB medium to an OD$_{600}$ of 0.1. Cultures were grown to OD$_{600}$ of 0.8 to 1.2 at 37 °C for induction of expression. Protein expression was induced by adding IPTG to a final concentration of 0.1-0.5 mM, and the expression temperature was set to 18 °C for 16 h (GST-CTD$_{[52]}$, GST-Cdk2) or 30 °C for 4 h (GST-CTD$_{[9]}$, GST-SPT5).

Bacteria were collected in 1 L buckets by centrifugation at $6200 \times g$ (JLA8.1 rotor, Beckman-Coulter) for 20 min. Cell pellets were resuspended in phosphate-buffered saline (PBS) and pelleted again by centrifugation. Bacterial pellets were subjected to cell lysis or snap-frozen in liquid nitrogen and stored at −20 °C for later use.

### Baculovirus generation

*Sf*9 cells were transfected with bacmid DNA by lipofection using Cellfectin (Invitrogen). Transfection was performed in 6-well format with 2 ml ($0.35 \times 10^6$ cells/ml) *Sf*9 cells per well. For transfection, 10 μl of Bacmid DNA was mixed with 100 μl serum-free medium. In a different tube, 8 μl Cellfectin was mixed with 100 μl medium. Both solutions were mixed and incubated for 15–30 minutes to allow formation of DNA-lipid complexes. After incubation, 200 μl was added to the respective well. Cells were incubated for 72 h at 27 °C. After 72 h the virus-containing supernatant was collected (V$_0$) and sterile filtered. V$_0$ was then used for virus amplification. V$_0$ was used to infect 50 ml of *Sf*9 cells ($0.5 \times 10^6$ cells/ml). Successful infection of the *Sf*9 cells was monitored by a cessation of cell division of transfected cells due to viral infection. Therefore, cells were counted every day and adjusted to $0.5 \times 10^6$ cells/ml (50 ml) until the cells stopped dividing. After replication had stopped, cells were incubated for another 48 h and then centrifuged at $500 \times g$ for 20 min. The supernatant containing the virus was collected (V$_1$). To obtain a higher titer virus (V$_2$), 1 ml of V$_1$ was used to infect 100 ml *Sf*9 cells ($1 \times 10^6$ cells/ml). Cells were incubated for four days and afterwards centrifuged at $500 \times g$ for 20 min and supernatant was collected (V$_2$). All viral stocks were sterile-filtered and stored at 4 °C. Expression was induced by infection of *Sf*9 cells at a density of $1.5 \times 10^6$ cells/ml with 2% of the V2 preparation and incubated for 72 h.

### Protein purification

For purification, cell pellets were resuspended in the respective lysis buffer and lysed by sonication. Cell debris was removed by centrifugation at $75,000 \times g$ in a JA25.50 rotor for 30 min at 4 °C. The lysate was afterwards filtered through a syringe filter with a 0.45 μm pore size. Filtered lysate was then used for affinity chromatography.

### Cdk7/Cyclin H complexes

Human GST-Cdk7 (2-346)/Cyclin H (1-323) was expressed from a single vector using the multibac$^{turbo}$ system. Cells were lysed in lysis buffer (50 mM Hepes pH 7.6, 150 mM NaCl, 5 mM β-mercaptoethanol). After filtration, lysate was applied to GSTrap 4FF affinity columns (Cytiva) using an ÄKTA FPLC system (Cytiva). Columns were washed extensively with lysis buffer and the protein complex eluted with elution buffer (20 mM HEPES pH 7.6, 150 mM NaCl, 1 mM tris(2-carboxyethyl) phosphine (TCEP), 10 mM reduced glutathione (GSH)). The GST-tag was removed by TEV protease digestion and the complex further purified by size exclusion chromatography (SEC) on a HiLoad 16/600 Superdex200 pg column (Cytiva) equilibrated with SEC buffer (20 mM HEPES pH 7.6, 150 mM NaCl, 1 mM TCEP) followed by reverse GST purification to remove residual GST and non-cleaved complexes.

### Cdk7/Cyclin H/Mat1 complexes (non-phosphorylated)

For generation of non-phosphorylated Cdk7/Cyclin H/Mat1 complexes, human GST-Cdk7 (2-346), Cyclin H (1-323) and Mat1 (either 1-309 or 230-309) were co-expressed in *Sf*9 cells from a single vector using the multibac$^{turbo}$ system. Cdk7/Cyclin H/Mat1 complexes were purified by GST affinity chromatography followed by SEC (SEC buffer: 20 mM HEPES pH 7.6, 150 mM NaCl, 1 mM TCEP) and reverse GST purification to remove residual tags and non-cleaved complexes.

### Cdk7/Cyclin H/Mat1 complexes (doubly phosphorylated)

For generation of phosphorylated Cdk7/Cyclin H/Mat1 complexes, human GST-Cdk7 (2-346) and Cyclin H (1-323) were co-expressed from a single vector using the multibac$^{turbo}$ system. A C-terminal construct of Mat1 fused to an N-terminal MBP, MBP-Mat1 (230-309), was expressed separately from Cdk7/Cyclin H in *Sf*9 cells. For purification of ternary CAK complexes, cell lysates of the individual GST-Cdk7/Cyclin H and MBP-Mat1 (230-309) expressions were pooled prior to purification. Cells were lysed in lysis buffer (50 mM Hepes pH 7.6, 150 mM NaCl, 5 mM β-mercaptoethanol). Complexes were affinity purified using MBPTrap HP columns (Cytiva) using an ÄKTA FPLC system (Cytiva). The columns were washed with lysis buffer until a stable baseline of absorbance was measured in the effluent. Proteins were eluted with lysis buffer containing 10 mM maltose. GST and MBP tags were removed by TEV protease digestion. The complex was further purified by SEC on a HiLoad 16/600 Superdex 200 pg column (Cytiva) equilibrated with SEC buffer (20 mM Hepes pH 7.6, 150 mM NaCl, 1 mM TCEP) and reverse MBP and GST affinity chromatography.

### Protein purification of substrate proteins

A DNA fragment encoding all 52 heptad repeats of the wild-type human RNAPII subunit Rpb1 CTD (residues 1587-1970, UniProt accession number P24928) was purchased from BioScience, UK (clone RPCIB753H14141Q), and cloned using *Nco*I/*Eco*RI restriction sites into a modified pGEX-6P1 expression vector, containing an N-terminal GST affinity tag followed by a PreScission 3C protease cleavage site. Protein

purification was carried out as described above, except that the GST-tag was not cleaved and the GST-CTD[52] protein was kept intact.

A cDNA encoding wild-type human SPT5 protein (residues 748-1087, UniProt accession number O00267) was sub-cloned from a cDNA library (GenBank accession code DQ896795) into a pET-28a expression vector modified with an N-terminal GST affinity tag followed by a TEV protease cleavage site using *Eco*RI/*Not*I restriction sites. Purification of SPT5 was carried out as described above for GST-fusion proteins using a preparative HiLoad 16/60 Superdex 75 column (Cytiva) for gel filtration. Protease cleavage was not performed and instead recombinant protein was kept as intact GST-SPT5.

An expression plasmid of human GST-Cdk2 was purchased from AddGene (plasmid #61845) and used without further subcloning. Protein was expressed and purified as described for GST-SPT5.

### Tag cleavage by TEV protease digestion
TEV protease digestion was performed at a TEV:protein ratio of 1:50 overnight at 4 °C.

### Synthetic CTD peptide substrates
Synthetic CTD peptides were purchased at HPLC grade quality (>95% purity) as customized synthesis from Biosyntan, Berlin.

### Nanobody generation, expression, and purification
Nanobodies were generated by the Core Facility Nanobodies, University Clinics Bonn, by repeated immunization of an alpaca with doubly T-loop phosphorylated Cdk7/Cyclin H/Mat1[230-309]. After immunization, peripheral blood mononuclear cells (PBMCs) were isolated, the mRNA extracted and reverse-transcribed to cDNA. VHH sequences were amplified by PCR using specific primers and cloned into a phagemid vector for phage display. Cdk7/Cyclin H/Mat1-binding VHH were enriched by phage panning with biotinylated target complex and screened by ELISA.

Nanobodies for co-crystallization with Cdk7/Cyclin H/Mat1 were expressed in *E. coli* BL21 (DE3) cells from a modified pET28a vector. The expression construct contained an N-terminal pelB sequence for periplasmic translocation followed by a TEV protease-cleavable hexahistidine tag for purification (pelB-His6-TEV-NB). Nanobodies for biochemical studies were expressed in *E. coli* WK6 cells from a pHEN6 vector with N-terminal pelB sequence and C-terminal HA-His-tag (pelB-NB-HA-His6). Pre-cultures were prepared as described above and diluted into larger volumes of TB medium to an $OD_{600}$ of 0.1. Cultures were grown to $OD_{600}$ of 1.0 at 37 °C for induction of expression. Protein expression was induced by adding IPTG to a final concentration of 1 mM, and the expression temperature was set to 30 °C for 16 h. Bacteria were collected in 1-L buckets by centrifugation at $6200 \times g$ (JLA8.1 rotor, Beckman-Coulter) for 20 minutes. Bacterial pellets were subjected to periplasmic extraction of proteins or snap-frozen in liquid nitrogen and stored at −20 °C for later use.

For purification, cells were resuspended in TES buffer (200 mM Tris pH 8.0, 0.65 mM EDTA, 500 mM sucrose) and incubated for 6 h at 4 °C. Periplasmic extracts were generated by osmotic shock in 0.25x TES buffer for 16 h at 4 °C. The extracts were cleared by centrifugation at $7720 \times g$ (JA25.50 rotor, Beckman-Coulter) for 45 min at 4 °C, filtered through syringe filter with 0.45 μm pore size, and subjected to affinity chromatography. The lysates were applied to HisTrap FF affinity columns (Cytiva) using an ÄKTA FPLC system (Cytiva). Columns were washed extensively with wash buffer (50 mM Tris pH 7.5, 150 mM NaCl, 10 mM imidazole) and eluted with wash buffer containing 500 mM imidazole. For crystallization, nanobodies were dialyzed against SEC buffer (20 mM HEPES pH 7.6, 150 mM NaCl) and the N-terminal His6-tag was removed by TEV protease digestion. The proteins were further purified by SEC on a HiLoad 16/600 Superdex75 pg column (Cytiva) equilibrated with SEC buffer, followed by reverse $Ni^{2+}$-NTA purification to remove non-cleaved protein. For biochemical assays, nanobodies

were affinity-purified as described above and further purified by SEC on a HiLoad 16/600 Superdex75 pg column (Cytiva) equilibrated with SEC buffer (20 mM HEPES pH 7.6, 150 mM NaCl).

### Crystallization and diffraction data collection
Initial crystallization screens of Cdk7/Cyclin H/Mat1/VHH[RD7-04] were set up using a homemade 96-well assay (0.1 M Hepes pH 7.0, 15% PEG 4 K). Purified CAK complex was concentrated to 14.7 mg/ml and mixed with 1 mM $ADP/Mg^{2+}$. The complex of CAK and nanobody was obtained by mixing Cdk7/Cyclin H/Mat1 and VHH[RD7-04] in a 1:1 ratio, followed by incubation on ice for 30 min prior to crystallization.

Hanging drops were set using 1:1 ratio of protein mixture and mother-liquor. Optimized rod-shaped crystals at approximate size of $50 \times 70 \times 90$ μm appeared in about three to five days at 15 °C in drops. The reservoir solution contained 0.1 M Hepes (pH 7.0), 12% (v/w) medium weight PEG mix of PEG 6K and PEG 4K, 10% ethylene-glycol, and 0.2 M non-detergent sulfobetaine (NDSB-201). Diffraction data of the samples were collected from a single loop-mounted crystal, each held in a gas stream of evaporating liquid nitrogen at a temperature of 100°K. The diffraction data sets (Supplementary Table 1) were collected at the P13 synchrotron beamline at Deutsches-Elektronen-Synchrotron (DESY) Hamburg, Germany, equipped with an Eiger detector.

### Structure determination and model building
Data were processed and scaled using the XDS program package[61]. The phase problem was solved by molecular replacement using PHASER[62]. The coordinates of the CAK from cryoEM (PDB 6XBZ)[25] were used as search models for the structure of Cdk7/Cyclin H/Mat1/VHH[RD7-04]. The model was refined by alternating cycles using PHENIX[63]. Manual rebuilding and visual comparisons were made using the graphical program COOT[64]. The stereochemical quality of the model was confirmed using a Ramachandran plot. The final structure includes two Cdk7/Cyclin H/Mat1/VHH[RD7-04] complexes that superimpose well with an RMSD value of 0.282 Å over 624 atoms, indicating the homogeneity of the complex formation. All protein could be built as a continuous chain without a single gap. A total of 384 water molecules and 6 ethylene glycol ligands were built, but no ADP was present in the nucleotide-binding sites of the kinases. The structural model at 2.15 Å resolution has been refined to $R_{work}$ and $R_{free}$ values of 20.3% and 23.8%, respectively. Details of the diffraction data collection, structure quality, and refinement statistics are given in Supplemental Table 1. Molecular diagrams were drawn using the PyMOL molecular graphics suite.

### Kinase assays
Kinase activity assays with recombinant proteins were performed in kinase assay buffer (50 mM Hepes pH 7.6, 34 mM KCl, 7 mM $MgCl_2$, 5 mM β-glycerophosphate, 2.5 mM DTE). If not indicated otherwise, reactions contained 0.1 μM kinase, were started by the addition of ATP to 1 mM final concentration, and incubated at 30 °C in a shaking incubator. For detection of the phosphorylation by a gel shift in SDS-PAGE or immunoblot analysis, the reactions were stopped by adding an equal volume of 2xSDS sample buffer. In all other cases, reactions were quenched by the addition of EDTA (25-50 mM final concentration).

For quantitative analysis, radioactive kinase assays were performed. Reactions were done in a total volume of 15 μL per sample. If not indicated otherwise, reactions contained 0.1 μM kinase, were started by the addition of ATP to 1 mM containing 0.35 μCi [32P]-γ-ATP (Perkin Elmer) and incubated at 30 °C in a shaking incubator. Reactions were stopped by adding EDTA to a final concentration of 50 mM. Samples were spotted onto Amersham Protran nitrocellulose membrane (GE Healthcare) filter sheets. Filters were washed three times for 5 min with PBS to remove free, non-reacted ATP. Samples were

transferred to 4-ml liquid scintillation vials and immersed in 2 mL liquid scintillator (UltimaGold). Radioactivity was counted in a Beckman Liquid Scintillation Counter (Beckman-Coulter) for 1 min.

## Data analysis and presentation

GraphPad Prism (v.7) was used for data analysis and initial data representation. Data were analyzed with in-built data analysis equations. $K_M$ values were determined by Michaelis-Menten equation $Y = V\max*X/(K_M + X)$. Time course reactions were fitted by non-linear curve fitting using a second-order polynomial equation $(Y = A + B*X + C*X^2)$. For linear regression, lines were forced to adhere to $X_0 = 0$. Figures were assembled using AffinityDesigner software.

## Immunoblotting

For immunoblotting, proteins were separated by SDS-PAGE and transferred to a nitrocellulose membrane (Optitran BA-S 85, pore size 0.45 μm, GE Healthcare) using a semi-dry blotting chamber at a constant current of 140 mA/gel for 60 minutes. After transfer, the membrane was blocked in 5% milk-powder in TBS-T for 1 h at room temperature or at 4 °C overnight. The membrane was incubated with a primary antibody.

The following antibodies were used in this study: α-pSer2 CTD, rat monoclonal, clone3E10, 1:100; α-pSer5 CTD, rat monoclonal, clone 3E8, 1:500; α-pSer7 CTD, rat monoclonal, clone 4E12, 1:100; α-pThr4 CTD, rat monoclonal, clone 1G7, 1:100 (CTD antibodies were a kind gift from Dirk Eick, Munich); rat monoclonal, clone 6D7, 1:1000 (Active-Motif #61362); α-Cdk7 (total), mouse monoclonal, clone 31TF2-1F8, 1:1000, (Invitrogen #MA3-001); α-Cdk7 (total), mouse monoclonal, clone C4, 1:1000 (Santa Cruz Biotechnology #sc-7344); α-phospho Cdk7 (pT170), rabbit polyclonal, 1:1000 (Affinity Biosciences #CPA5749); α-phospho-Cdk7 (pT170), rabbit polyclonal, 1:1000 [13]; α-phospho-Cdk7 (pS164), rabbit polyclonal, 1:1000 (Invitrogen #PA5-105583); α-GST, mouse monoclonal, 1:1000 (Thermo Fisher Scientific # 740007M); α-HA tag, mouse monoclonal, clone F-7 (Santa Cruz Biotechnology #sc-7392); α-MBP, rabbit polyclonal, 1:10 for immobilization on SPR sensor ship (Novus Biologicals, NBP2-22462). Blots were washed 3 × 5 minutes in TBS-T and then incubated with appropriate secondary antibody: goat anti-rat HRP, 1:100,000, (Cell Signaling Technology, #7077); chicken anti-rat HRP, 1:5000, (Santa Cruz Biotechnology #sc2956); goat anti-mouse IRDye 680RD, 1:10,000 (Licor); or donkey anti-rabbit IRDye 800CW, 1:10,000 (Licor) for 1 h. Membrane was washed 3 × 5 min in TBS-T. HRP-coupled antibodies were analyzed with a CCD camera in a XRSChemDoc system (BioRad) after immersion with ECL-solution (Sigma Aldrich) for 1 min. Infrared dye coupled secondary antibodies were analyzed in a Licor Odyssey Clx imager (Licor). Uncropped images of the blots are provided in the Source Data file accompanying this paper.

## Phostag-SDS PAGE

For analysis of Cdk7 phosphoisoforms from cell lysates, lysates were resolved on 10% SDS-PAGE gels containing 40 μM PhosTag-Acrylamide (Fujifilm Wako Chemicals) and 80 μM $MnCl_2$. Samples were transferred to nitrocellulose using transfer buffer (25 mM Tris-HCl, 192 mM glycine, pH 8.3, 20% (v/v) methanol, 0.1% SDS) in a tank-blot system (Biorad) at a constant current of 400 mA for 2 h at 4 °C. After blotting, immunoblotting was performed as described above.

## Cell culture and preparation of cell lysates

HCT116 cells were grown in DMEM medium supplemented with 10% FBS at 37 °C in a humidified atmosphere containing 5% $CO_2$. For preparation of cell lysates, cells at 70–80% confluency were washed twice with ice cold PBS and lysed by incubation in radioimmunoprecipitation assay (RIPA) buffer containing protease and phosphatase inhibitors for 5 min. Samples were then sonicated to ensure solubilization of chromatin-bound proteins. Lysates were cleared of residual debris by centrifugation at $15,000 \times g$ for 10 min at 4 °C. Total protein concentration was determined using a Pierce BCA assay (Thermo Fisher) according to the manufacturer's instructions. Samples were boiled in SDS sample buffer for 5 minutes and kept at −20 °C until used.

## Transfection of HCT116 cells

HCT116 cells were transfected with Cdk7-HA in a pcDNA5/FRT vector using lipofectamine3000 reagent (Invitrogen) according to the manufacturer's instructions. Cells were grown in 6-well plates to reach 70% confluency prior to transfection. Cells were lysed after 48 h as described above.

## Protein thermal stability analyses

Thermal stability of proteins was analyzed using nanoscale differential scanning fluorimetry with a Prometheus NT.48 (NanoTemper) device. Denaturation of proteins was monitored by changes in internal fluorescence at wavelengths of 330 and 350 nm. Proteins were diluted to 2.5 μM in storage buffer (20 mM Hepes, pH 7.6, 150 mM NaCl, 1 mM TCEP) and the thermal stability was monitored from 20 °C to 90 °C at a heating rate of 1.5 °C/min using the PR.ThermControl software (NanoTemper).

## Surface plasmon resonance measurements

Surface plasmon resonance (SPR) spectroscopy experiments were performed using a Biacore 8 K instrument (Cytiva). All steps were performed at 25 °C. The system was equilibrated with running buffer (10 mM HEPES pH 7.4, 150 mM NaCl, 0.05% Tween20). The anti-MBP antibody used to capture MBP-Mat1$_{230-309}$ was immobilized in running buffer using amine coupling. Before protein immobilization, flow cell surfaces of a CM5 sensor chip were activated for 15 s with 50 mM NaOH (30 μL/min), followed by activation with a 1:1 mixture of 0.1 M NHS (N-hydroxysuccinimide) and 0.1 M EDC (3-(N,N-dimethylamino) propyl-N-ethylcarbodiimide) (10 μL/min) for 7 min. The flow system was washed with 1 M ethanolamine pH 8.0. The anti-MBP antibody was diluted 1:10 in acetate buffer. Immobilization was carried out on the flow cell 1 surface for 160 s at a flow rate of 10 μL/min. Subsequently, surfaces were blocked with 1 M ethanolamine pH 8.0 (10 μL/min) for 7 min. MBP-Mat1 was immobilized at a concentration of 15 nM. Kinetic binding of Cdk-Cyclin complexes was measured as multi-cycle kinetics. The complexes were injected (30 μL/min, association: 180 s, dissociation: 900 s) at increasing concentrations of 0.6, 1.9, 5.6, 16.7, 50 and 150 nM. Data were collected at a rate of 10 Hz. The data were double-referenced by blank cycle and reference flow cell subtraction. Additionally, binding to MBP was excluded by an MBP control measurement. Processed data were fitted with steady state affinity binding determination using the Biacore Insight Evaluation Software (Cytiva).

## Mass spectrometry analyses (total intact mass determination)

Total intact mass was determined by ESI-MS at the mass spectrometry facility in St. Andrews.

## Statistics and reproducibility

No statistical methods were used to predetermine sample size. The experiments were not randomized and the investigators were not blinded to allocation during experiments and outcome assessment. Numbers of total repeats of representative images: Fig. 1a: $N = 2$; Fig. 3a: $N = 2$; Fig. 3c: $N = 2$; Fig. 4c: $N = 2$; Fig. 6a: $N = 2$; Fig. 6b: for wt, T170A, S164A $N = 3$, for S164A/T170A, S164E $N = 2$.

## Reporting summary

Further information on research design is available in the Nature Portfolio Reporting Summary linked to this article.

## Data availability

The authors declare that the data supporting the findings of this study are available within the paper and its supplementary information files. Source data are provided with this paper. Structure coordinates and diffraction data of the human Cdk7/Cyclin H/Mat1/VHH$_{RD7-04}$ complex were deposited in the Protein Data Bank (http://www.pdb.org) under accession codes 8PYR. The coordinate data used in this study are available in the PDB database under accession codes 6XBZ, 3QHR, 1UA2, and 8ORM. Source data are provided with this paper.

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

## Acknowledgements

We would like to thank Jale Sariyar, Melanie Specht, and Elif Tokmak for excellent technical assistance and Gregor Hagelueken for discussions and help with CTD modelling. We thank Sally Shirran, University of St. Andrews, for mass spectrometry analysis and Paul-Albert König and the Core Facility Nanobodies, University Clinics Bonn, for initial nanobody generation and selection. This work was supported by a grant from the Deutsche Krebshilfe (70114008) to M.G., by National Institutes of Health grant R35 GM127289 to R.P.F., and by a postdoctoral fellowship of the German Academic Exchange Service (DAAD) to R.D. (57584491). M.G. is funded by the Deutsche Forschungsgemeinschaft (DFG) under Germany's Excellence Strategy – EXC2151–390873048.

## Author contributions

R.D. expressed and purified proteins and performed biochemical experiments. K.A. crystallized proteins and determined the structure; S.B. expressed and purified nanobodies for structure determination; M.S. measured substrate specificities and K.G. performed SPR experiments. R.D., R.P.F. and M.G. conceptualized the study and wrote the manuscript. All authors contributed to the final version of the manuscript.

## Funding

## Competing interests

The authors declare no competing interests.
