## [Peer Review File · Nature Communications]

Structural basis of Cdk7 activation by dual T-loop phosphorylationREVIEWER COMMENTS

Reviewer #1 (Remarks to the Author):

In this manuscript, the authors have determined the structure of doubly phosphorylated Cdk7, a key activator of the cell cycle and a regulator of RNA polymerase II (RNAPII). Furthermore, based on their experimental work with human colon cancer cells, the authors suggest that Cdk7 activation may involve a two-step process. This is substantiated by biochemical analyses demonstrating that Cdk7, when doubly phosphorylated at the T-loop, exhibits increased activity towards multisite targets associated with transcription.

I think this manuscript offers valuable contributions to the cell cycle and transcription control fields by 1) providing structural basis of Cdk7 activation and 2) showing the relevance of dual phosphorylation in Cdk7 activity and substrate selection. Consequently, I am positive and consider this manuscript to meet the standards for publication in the journal Nature Communications. However, it is important to note that some improvements are necessary before I can provide my full support. Changes should be made to address the following concerns:

Major concerns

1. In Figure 3b, the signal for Cdk2 in the in vitro kinase assay appears significantly weaker compared to that for RNAPII CTD. It is crucial to confirm whether the total amount of CTD and Cdk2 used in the kinase assay was equal. Alternatively, could these results be expected due to the higher number of phosphorylatable residues in CTD compared to Cdk2? To bolster their argument regarding the specificity change of Cdk7 for RNAPII CTD versus Cdk2 after T-Loop phosphorylation, the authors should consider including a non-phosphorylatable mutant form of Cdk2 as a control. This control would help to demonstrate the level of background signal in the kinase assay.
2. In Figure 4 and Supplementary Figure 5, the authors claim that Cdk7 can phosphorylate the T4 residue of CTD. Since this is a novel claim, it should be shown more convincingly. To strengthen this finding, the authors should consider providing evidence regarding the specificity of the anti-pT4 antibody used in their experiments. Additionally, the authors could enhance their case by conducting an in vitro kinase assay using CTD with all but one phosphorylatable residue mutated to alanine. This approach would help demonstrate the site specificity of the phosphorylation reaction.
3. In Figure 4c, the authors should show whether the activity of the Cdk7 T-loop phosphorylated forms is the same for every position in the CTD. Especially, considering that phosphorylation of one residue excludes phosphorylation of others in the same repeat, different kinetics for different residues could lead to substantially different phosphorylation patterns. The authors could, for example, show this by using CTDs where all but one phosphorylatable residue have been mutated to alanine using shorter time points as shown in Figure 4A.
4. The presentation of protein gels supplemented with Phos-tag could benefit from improvements to enhance clarity. For better comprehension, the authors should consider indicating the migration positions of each phospho-form and distinguishing between endogenous and exogenous protein forms. Presently, the data in Figure 6 is challenging to interpret, even for individuals with extensive experience in analyzing Phos-tag gels and may pose difficulties for a broader readership.
5. Unfortunately, the authors encountered difficulty in distinguishing between the doubly phosphorylated and T170 mono-phosphorylated forms using the Phos-tag gels (Figure 6). In Figure 6a, it is noteworthy that the anti-pS164 antibody appears to recognize two additional slowly migrating proteins positioned above the signal for pT170. It would be valuable for the authors to clarify whether these bands are specific to Cdk7 and whether they may correspond to the double-phosphorylated form of the protein. Furthermore, to provide additional insight into the phosphorylation states, the authors should consider including a phosphatase control in Figure 6a. This control would illustrate the migration position of the non-phosphorylated protein, aiding in the interpretation of the results and adding confidence to their sequential phosphorylation model.

Minor concerns

1. The data shown in Supplementary Figure 5b suggests that Cdk9 cannot phosphorylate CTD at S2 position, a well-described target of Cdk9. The authors should show whether the used anti-pS2 antibody is functional, and if it is, elaborate why Cdk9 is unable to phosphorylate its known target

substrate.

2. Please indicate centrifugation speeds in g or include the used centrifuge rotor (lines 513 and 516).

3. Supplementary Figure 6c could be depicted using some other graph type. 3D graphs are generally not a good practice.

4. Line 280 is missing a reference to figure.

Reviewer #2 (Remarks to the Author):

This manuscript reports structural and biochemical analysis of activation loop phosphorylation in the kinase CDK7. CDK7 and its activators Cyclin H and Mat1 regulate the cell cycle, through phosphorylation of other cyclin dependent kinases, and transcription, through phosphorylation of the RNA polymerase II CTD and other targets. CDK7 has been well studied because of its critical role in these fundamental processes and as a drug target. However, it has remained unclear what the relative significance and function are of two phosphorylation sites in the activation loop (also called T loop for CDKs). In particular, the more noncanonical site S164 has uncertain function, and reports on its role in CDK7 regulation have described contrary results depending on the experimental and biological context. The study here is an effort to clarify the role of the two T loop sites in kinase activity. The authors determine a crystal structure of phosphorylated CDK7 in complex with Cyclin H and Mat1 using a clever purification strategy to obtain homogeneous sample. They use kinase assays with various substrates to understand the significance of the two phosphorylation events on activity, and they test the interdependence of the phosphorylation events in cells. The structural analysis is of high quality, and, with a few exceptions detailed below, conclusions are well supported by the results. There are several interesting findings that will be useful to the field, including how S164 phosphorylation coordinates and stabilizes the trimer and the implication of the site as having important regulatory function for CDK7 activity specifically in multisite phosphorylation. The impact of the study is likely somewhat limited considering it falls short of providing a detailed mechanistic understanding of how S164 phosphorylation impacts catalytic activity and providing strong support for the relevant biological context. Significant concerns and some minor points are detailed below.

1) The kinetic analysis needs to be more thorough and detailed. Activity assays that measure single substrate concentrations at long time points are insufficient. Proper steady state measurements at different substrate and ATP concentrations would be more robust, and they could be used to measure parameters such as K_M and k_{cat} that could provide more mechanistic insights.

2) Despite the need for more rigorous analysis, the biochemical data at least suggest that S164 phosphorylation is critical for Mat1 stimulation of kinase activity. However, there is still no clear explanation for why this is the case. The ternary complex still forms without this phosphorylation and SPR measurements here show affinity is similar. While there are some thermostability differences, it is not obvious how relevant these differences are to activity differences that are measured well below the melting temperatures. The impact of T loop phosphorylation on substrate and ATP binding and the catalytic turnover is not explored. As already described, more detailed steady state kinetic analysis may shed some insights. Also, a better description of how the phosphorylated structure compares to the unphosphorylated structures previously published is needed and may provide some explanation into how T loop phosphorylation is structurally coupled to the substrate binding and active sites.

3) Similarly there is no mechanistic or structural explanation for why CDK7 T loop phosphorylation stimulates multisite phosphorylation in the CTD. There are some hypotheses proposed in the discussion but no experimental data, and as presented, the results in Fig. 4 are interesting but superficial.

4) Because the authors only show relative activity in Fig. 4d, it is unclear how activity of the enzyme compares between substrates. For example, it is unclear to what extent unphosphorylated dimer or trimer activity varies by substrate, and so the contribution of Mat1 to multisite phosphorylation is not evident in the data. The authors should show activities here that are not normalized.

5) The conclusion on line 198 that "Mutation of Cdk7 T170 to alanine reduced the levels of S164 phosphorylation, whereas the Cdk7 S164A mutation had no effect on pT170 levels" is based on comparison of band intensities in a Western blot. The differences seem subtle, and there is no indication of replicates. A more quantitative assay such as mass spectrometry would better support the conclusion.

6) The conclusion on line 200 states that "Mutation of Cdk7 T170 severely reduced activity to levels close to those of binary Cdk7/Cyclin H complexes," however, data for the binary complex are not shown in Fig. 3b. Perhaps authors are referring to data in Fig. 3d? Similarly, beginning on line 206, a comparison is made to unphosphorylated CDK7, but it is not clear where those data can be found.

7) A few suggestions for the discussion: The first few paragraphs largely recapitulate structural results and could be condensed or removed. The logic leading to the conclusion in the last sentence is unclear. It may be interesting to discuss the implications of the results for pharmacological inhibition of CDK7.

Reviewer #3 (Remarks to the Author):

Düster et al. present a CAK crystal structure of a CAK that is completely phosphorylated at the T-loop positions S164 and T170. The 2.15 Å crystal structure was accomplished by diligent and clever usage of nanobodies. The structural data are supplemented with a detailed biochemical analysis concerning different phosphorylation states of the CDK7 T-loop and the effects thereof on RNAP II CTD, SPT5, and CDK2 phosphorylation. The authors can show that CAK with a fully phosphorylated T-loop has the highest activity towards the CTD, whereas CDK2 phosphorylation is almost unaffected. The authors discuss in detail the structural necessities and outcomes of the phosphorylation events in terms of activity.

Although the work is interesting it is important to address several concerns prior to publication.

The biochemical analysis of the complex is very interesting and compelling. However, the authors have not provided a real ground state for their activity measurements. T-loop phosphorylated CDK7/Cyclin H is compared with non-phosphorylated trimeric CAK or fully phosphorylated trimeric CAK that was obtained by mixing lysates but the analysis of unphosphorylated CDK7/Cyclin H is missing to provide a ground state. The T-loop phosphorylated CDK7/Cyclin H displays already significant activity that is hardly increased by the presence of MAT1 for the non phosphorylated complex. Previous data indicate, however, that CDK7 or CDK7/Cyclin H activity is significantly lower than CAK activity (Lolli 2004, Peisert 2020,) in the non-phosphorylated state. It is thus important to provide data for the ground state to obtain a complete picture of the phosphorylation effects on CAK activity.

A serious concern is the description of the CAK structure. The authors mention previously obtained CAK structures but fail to include a detailed comparison in a figure, with special emphasis on T-loop movement and positioning. A detailed comparison would have shown that the T-loop position between all available structures is highly similar. This analysis should be clearly included in the data interpretation especially in the paragraph "dual phosphorylation puts the T-loop in its place" which at least adds a question mark to the subtitle. In fact, the similarity of the structures raises the question why the activity is altered although the T-loop positioning is structurally very similar. The different interactions of the phosphate networks are appreciated. However, in a superposition with the coordinates of human CAK (pdb entry 8orm) of seemingly non phosphorylated CAK the side-chain positions do not seem to differ significantly for the T-loop and the phosphate receiving residues. This should be considered and discussed accordingly. It is also important to mention that the missing first 45 residues of the N-lobe of CDK7 constitute a major part of the ATP binding pocket that is virtually not present in the here presented structure.

An additional concern is that the biochemical data sometimes seems to be rather inconsistent between different experiments. In Figure 1c the activation of CDK7/Cyclin H by the addition of MAT1 is approx. 4-5 fold, whereas it is 8-10 fold in panels 3d and 3f. The main difference between these experiments seems to be that MAT1 was added later and in 4 fold excess of CDK7 (for the results shown in Figure 3d and 3f). The elevated activity of these complexes could indicate that the initial purification strategy that has also been applied in other experiments in the manuscript, leads to less and unpredictable activity patterns. In Fig. 4a, however, the difference at 15 min seems to

be tenfold or maybe even more with the copurification strategy. To overcome this high variability the authors should stick to one method of complex preparation for their biochemistry or clearly state and discuss differences between the approaches. In addition, it seems that the stoichiometry of the complexes presented in panel 3a significantly differs and thus cannot be compared as such. It could be necessary to pursue a different purification strategy? If possible, biological duplicates for every key experiment should be performed, rather than technical replicates especially in light of the issues to obtain an active stoichiometric complex. Overall, the different range of data points for many experiments seems odd. The aim should be to perform most of the experiments with a similar amount of technical and biological replicates.

The authors may consider to show the concentration dependent activity of the analyzed protein complexes since the enzyme to substrate ratio used here is 1:100 which is rather high on the substrate side. A complementary method like a coupled ATPase assay would provide more kinetic data than the 1 point measurements used here and also be helpful to assess the quality of the obtained data.

Finally, the discussion section should address differences and similarities to previous structures especially the T-loop position and the phosphate networks.

Minor issues:

In panel 6a a more annotation of the different bands would help the reader to understand the meaning of the figure.

In panel 2b all side chains of amino acids are visible except the one of F162. For consistency this should also be shown.

When the effect of pre-existing phosphorylation on the kinase activity was investigated (panel 4e and 5c in the supplements), the variant pS5-mid was omitted without reason. Furthermore, background information of why Ser7 was exchanged to a lysine would be helpful.

Please, correct the typo in the caption of panel 2f in the supplements since the crystal structure was obtained using VHH(RD7-04) instead of VHH(RD7-01), as correctly shown in the figure itself.

Detailed point-to-point reply to the Reviewers' comments:

Reviewer #1 (Remarks to the Author):

In this manuscript, the authors have determined the structure of doubly phosphorylated Cdk7, a key activator of the cell cycle and a regulator of RNA polymerase II (RNAPII). Furthermore, based on their experimental work with human colon cancer cells, the authors suggest that Cdk7 activation may involve a two-step process. This is substantiated by biochemical analyses demonstrating that Cdk7, when doubly phosphorylated at the T-loop, exhibits increased activity towards multisite targets associated with transcription.

I think this manuscript offers valuable contributions to the cell cycle and transcription control fields by 1) providing structural basis of Cdk7 activation and 2) showing the relevance of dual phosphorylation in Cdk7 activity and substrate selection. Consequently, I am positive and consider this manuscript to meet the standards for publication in the journal Nature Communications. However, it is important to note that some improvements are necessary before I can provide my full support. Changes should be made to address the following concerns:

We thank the Reviewer for the careful assessment of our study. We hope that we have adequately addressed all the concerns raised. A detailed point-by-point response is provided below.

Major concerns

1. In Figure 3b, the signal for Cdk2 in the *in vitro* kinase assay appears significantly weaker compared to that for RNAPII CTD. It is crucial to confirm whether the total amount of CTD and Cdk2 used in the kinase assay was equal. Alternatively, could these results be expected due to the higher number of phosphorylatable residues in CTD compared to Cdk2? To bolster their argument regarding the specificity change of Cdk7 for RNAPII CTD versus Cdk2 after T-Loop phosphorylation, the authors should consider including a non-phosphorylatable mutant form of Cdk2 as a control. This control would help to demonstrate the level of background signal in the kinase assay.

The phosphorylation signal of Cdk2 is indeed significantly weaker compared to CTD. In fact, CTD is a very powerful *in vitro* substrate for Cdk7. Both substrates, GST-CTD and GST-Cdk2 were purified to homogeneity and used at similar concentrations of 10 μ M GST-CTD_[52] and 15 μ M GST-Cdk2. The difference in signal strength originates from the smaller number of phosphorylation sites in the Cdk2 sample and the slower reaction kinetics, as previously documented in ref. 20 (Larochelle et al., EMBO J. 2001).

To address this issue experimentally, we included time course measurements of Cdk2 phosphorylation by three Cdk7 preparations (S_T, A_pT and pS_pT) in the revised manuscript (new Supplementary Figure 5h). The Cdk7 preparations display no significant differences in Cdk2 phosphorylation. After 30 min, the time point used for single time point measurements within this study, the reaction is within the linear range. We also followed the Reviewer's suggestion and included a Cdk2 T160A control to confirm that the signal represents indeed the T160 phosphorylation of Cdk2 within the T loop. We provide these data in the new Supplementary Figure 5i.

2. In Figure 4 and Supplementary Figure 5, the authors claim that Cdk7 can phosphorylate the T4 residue of CTD. Since this is a novel claim, it should be shown more convincingly. To strengthen this finding, the authors should consider providing evidence regarding the specificity of the anti-pT4 antibody used in their experiments. Additionally, the authors could enhance their case by conducting an *in vitro* kinase assay using CTD with all but one phosphorylatable residue mutated to alanine. This approach would help demonstrate the site specificity of the phosphorylation reaction.

We agree with the Reviewer that phosphorylation of Thr4 within the CTD requires a deeper analysis. However, we think that an 'all-but-one' mutational strategy would provide only limited insight, due to the highly artificial nature of the resulting substrate (repeat sequence: FAPTAPA). To bolster our analysis, we repeated the experiment using a different monoclonal pT4 antibody showing the same result (new Supplementary Figure 5e). Additionally, we want to refer the Reviewer to the P-TEFb (Cdk9/Cyclin T1) control that was included in the analysis. *In vitro*, P-TEFb phosphorylates mainly Ser5 and Ser7 residues of the CTD, as Cdk7 does. Even if we cannot exclude specific CTD phosphorylation patterns resulting in mis-identification, the absence of pT4 signal in the P-TEFb treated sample argues against off target binding of the tested antibodies.

3. In Figure 4c, the authors should show whether the activity of the Cdk7 T-loop phosphorylated forms is the same for every position in the CTD. Especially, considering that phosphorylation of one residue excludes phosphorylation of others in the same repeat, different kinetics for different residues could lead to substantially different phosphorylation patterns. The authors could, for example, show this by using CTDs where all but one phosphorylatable residue have been mutated to alanine using shorter time points as shown in Figure 4A.

We appreciate the Reviewer's interest in the CTD code and potential combinations of modifications generated by the different Cdk7 forms. In our view an informative experimental setup would require the use of full-length RNAPII CTD, as peptide substrates often do not recapitulate the actual substrate preferences. Regarding the use of an 'all-but-one' mutant CTD, please refer to our concerns stated above in response to point 2. Notably, our laboratories were among the first to use pre-phosphorylated synthetic CTD peptides to analyse with recombinant kinases the specificity of CTD recognition, the preferred phosphorylation sites, and possible dual phosphorylation patterns (PMID: 22588304; PMID: 24662513; PMID: 26748711). However, the use of short peptide substrates provides only limited insights into the direct substrate recognition motifs while a more comprehensive analysis would require at least eight heptad repeats to cover the full recognition surface of the CAK complex (see Supplementary Figure 8, where 6 repeats are required to bridge from the kinase catalytic site to the RxL-compatible recognition site on the cyclin subunit). While this consideration complies with the extended and repetitive nature of the RNAPII CTD, it is currently beyond a feasible approach for synthetic CTD peptides with either mutational or preset phosphorylation patterns.

4. The presentation of protein gels supplemented with Phos-tag could benefit from improvements to enhance clarity. For better comprehension, the authors should consider indicating the migration positions of each phospho-form and distinguishing between endogenous and exogenous protein forms. Presently, the data in Figure 6 is challenging to interpret, even for individuals with extensive experience in analyzing Phos-tag gels and may pose difficulties for a broader readership.

We revised the main Figure 6 for clarity. This includes labelling of the bands as well as reducing data complexity. In addition to labelling the gels with the respective phospho-isoform we provide a phostag gel with recombinant Cdk7, wt and T-loop mutants, as reference in the new Supplementary Figure 7a. We hope the Reviewer finds our changes helpful.

5. Unfortunately, the authors encountered difficulty in distinguishing between the doubly phosphorylated and T170 mono-phosphorylated forms using the Phos-tag gels (Figure 6). In Figure 6a, it is noteworthy that the anti-pS164 antibody appears to recognize two additional slowly migrating proteins positioned above the signal for pT170. It would be valuable for the authors to clarify whether these bands are specific to Cdk7 and whether they may correspond to the double-phosphorylated form of the protein. Furthermore, to provide additional insight into the phosphorylation states, the authors should consider including a phosphatase control in Figure 6a. This control would illustrate the migration position of the non-phosphorylated protein, aiding in the interpretation of the results and adding confidence to their sequential phosphorylation model.

We thank the Reviewer for the valuable suggestions and realized our data presentation needs clarification. In fact, we are able to separate pT170 phosphorylated Cdk7 from doubly phosphorylated pS164/pT170 Cdk7 in our phostag approach. We now provide a reference gel using recombinant Cdk7 and the respective mutants to illustrate the migration behaviour in phostag supplemented gels in the new Supplementary Figure 7a. A band corresponding to a Cdk7 isoform singly phosphorylated at T170 was not detected in HCT116 cells. This holds true for analysis of both endogenous Cdk7 and for transfected Cdk7-HA (wt).

Regarding the additional, slowly migrating bands recognized by the anti-pSer164 antibody it is important to note that the pSer164 antibody displays several off-target signals in conventional SDS-WB experiments. Migration behaviour of these bands cannot be predicted in phostag gels. Together with the fact that they were identified neither by the pT170 antibody nor by the total Cdk7 antibody, these bands almost certainly result from off-target binding of the antibody. The non-specific cross-reactivity of this antibody is not surprising given that the phospho-epitope against which it was raised, pS-P-N-R, is a perfect match to the consensus recognition site for CDKs (in contrast to pT170). We attempted to de-phosphorylate endogenous Cdk7 after immunoprecipitation but could not detect any decrease in pT170 phosphorylation. This resistance to phosphatase treatment has been described before (Labbé and Martinez et al., PMID: 7957080; Akoulitchev and Reinberg PMID: 9832506).

Minor concerns

1. The data shown in Supplementary Figure 5b suggests that Cdk9 cannot phosphorylate CTD at S2 position, a well-described target of Cdk9. The authors should show whether the used anti-pS2 antibody is functional, and if it is, elaborate why Cdk9 is unable to phosphorylate its known target substrate.

We thank the Reviewer for pointing to this discrepancy with what is commonly described in the literature. There is indeed a difference between Cdk9's substrate specificity as determined with recombinant protein and the effects on CTD phosphorylation obtained with CDK inhibitors such as flavopiridol *in vivo*. *In vitro*, Cdk9 primarily phosphorylates Ser5 as described by Czudnochowski et al. (2012; PMID: 22588304) and reported also by other groups (PMID: 28497798; PMID: 25561469). We also note that more selective inhibitors of Cdk9, such as NVP-2, have much weaker effects on Ser2 phosphorylation in human cells, especially when shorter incubation times are used to limit indirect effects (Parua et al., 2020 PMID: 32859893). One study by the Zhang group suggests that the ability to phosphorylate Ser2 depends on priming by a pY1 in the adjacent repeat (PMID: 32568517). We now address this in the revised version of the manuscript and refer to the relevant literature. The antibodies used in our manuscript are from Dirk Eick and Elisabeth Kremmer and used in many other studies. Notably, the same set of antibodies was used in our study of DYRK1A and HIP kinases, where we observed Ser2 phosphorylation signals (PMID: 34785661).

2. Please indicate centrifugation speeds in g or include the used centrifuge rotor (lines 513 and 516).

Thanks, corrected.

3. Supplementary Figure 6c could be depicted using some other graph type. 3D graphs are generally not a good practice.

Because the data in Fig. S6c are redundant with Fig. 5d we removed the panel in the revised figure.

4. Line 280 is missing a reference to figure.

Thanks, corrected.

Reviewer #2 (Remarks to the Author):

This manuscript reports structural and biochemical analysis of activation loop phosphorylation in the kinase CDK7. CDK7 and its activators Cyclin H and Mat1 regulate the cell cycle, through phosphorylation of other cyclin dependent kinases, and transcription, through phosphorylation of the RNA polymerase II CTD and other targets. CDK7 has been well studied because of its critical role in these fundamental processes and as a drug target. However, it has remained unclear what the relative significance and function are of two phosphorylation sites in the activation loop (also called T loop for CDKs). In particular, the more noncanonical site S164 has uncertain function, and reports on its role in CDK7 regulation have described contrary results depending on the experimental and biological context. The study here is an effort to

clarify the role of the two T loop sites in kinase activity. The authors determine a crystal structure of phosphorylated CDK7 in complex with Cyclin H and Mat1 using a clever purification strategy to obtain homogeneous sample. They use kinase assays with various substrates to understand the significance of the two phosphorylation events on activity, and they test the interdependence of the phosphorylation events in cells. The structural analysis is of high quality, and, with a few exceptions detailed below, conclusions are well supported by the results. There are several interesting findings that will be useful to the field, including how S164 phosphorylation coordinates and stabilizes the trimer and the implication of the site as having important regulatory function for CDK7 activity specifically in multisite phosphorylation. The impact of the study is likely somewhat limited considering it falls short of providing a detailed mechanistic understanding of how S164 phosphorylation impacts catalytic activity and providing strong support for the relevant biological context. Significant concerns and some minor points are detailed below.

1) The kinetic analysis needs to be more thorough and detailed. Activity assays that measure single substrate concentrations at long time points are insufficient. Proper steady state measurements at different substrate and ATP concentrations would be more robust, and they could be used to measure parameters such as K_M and k_{cat} that could provide more mechanistic insights.

To address the Reviewer's concerns, we determined the K_M for CTD and ATP for different Cdk7 preparations. We measured the kinase activity of three differently phosphorylated Cdk7 preparations (S_T, A_pT and pS_pT) at different CTD concentrations after 5, 10, and 20 minutes. For determination of the K_M for CTD we used the 10 min time point data at which the reaction is in the linear range. Although the difference in activity is evident in these measurements, there is no difference regarding the K_M for CTD. For ATP, we determined highly similar K_M values of: S_T, 51.7 μ M; A_pT, 40.3 μ M; pS_pT, 75.5 μ M. These values are well below the 1 mM ATP used throughout the study. The change in activity hence relies in a change of k_{cat} but not K_M . The new data are presented in the new Supplementary Figure 5a-c.

2) Despite the need for more rigorous analysis, the biochemical data at least suggest that S164 phosphorylation is critical for Mat1 stimulation of kinase activity. However, there is still no clear explanation for why this is the case. The ternary complex still forms without this phosphorylation and SPR measurements here show affinity is similar. While there are some thermostability differences, it is not obvious how relevant these differences are to activity differences that are measured well below the melting temperatures. The impact of T loop phosphorylation on substrate and ATP binding and the catalytic turnover is not explored. As already described, more detailed steady state kinetic analysis may shed some insights. Also, a better description of how the phosphorylated structure compares to the unphosphorylated structures previously published is needed and may provide some explanation into how T loop phosphorylation is structurally coupled to the substrate binding and active sites.

We followed the Reviewer's suggestion and compared our doubly phosphorylated structure with the unphosphorylated or singly T170 phosphorylated Cdk7 structure in the new Figure 7 of the revised manuscript. This comparison shows that the electrostatic surface potential is significantly different between the unphosphorylated Cdk7/Cyclin H/Mat1 complex and the doubly phosphorylated CAK complex. With respect to the comments of all Reviewers, this

comparison to existing structures is now described in the first two paragraphs of the revised Discussion. We also determined the impact of T loop phosphorylation on ATP binding and CTD phosphorylation (new Supplementary Fig. 5a–c).

3) Similarly, there is no mechanistic or structural explanation for why CDK7 T loop phosphorylation stimulates multisite phosphorylation in the CTD. There are some hypotheses proposed in the discussion but no experimental data, and as presented, the results in Fig. 4 are interesting but superficial.

We agree with the Reviewer that our understanding of the mechanism by which Cdk7 T loop phosphorylation selectively stimulates CTD kinase activity is incomplete. However, in light of the fact that none of the four recent manuscripts describing Cdk7/CycH/Mat1 structures from various species (Greber et al., PNAS, 2020; Peissert et al., PNAS, 2020; van Eeuwen et al., Sci. Adv., 2021; Cushing et al., Nat. Commun., 2024) contain functional analysis or kinetic CTD (or Cdk2) substrate phosphorylation data, we do not agree that our study can be characterized as superficial. As we show here, the double T-loop phosphorylations significantly change the electrostatic characteristics of the Cdk7 kinase from a largely basic surface to a more neutral one (Fig. 7c,d). This patch is more than 15 Å away from the kinase catalytic centre and known to be involved in substrate recognition as shown by the work of Louise Johnson and others (Brown et al., Nat. Cell. Biol., 1999; Bao et al., Structure, 2011; Abdella et al., Science, 2021). For an understanding of the multisite phosphorylation of the CTD and the mechanism of processivity and directionality of phosphorylation as well as the recognition of pre-set phosphorylation marks it would be great to have complex structures of the CAK complex with substrate. This is however beyond the scope of the present study.

4) Because the authors only show relative activity in Fig. 4d, it is unclear how activity of the enzyme compares between substrates. For example, it is unclear to what extent unphosphorylated dimer or trimer activity varies by substrate, and so the contribution of Mat1 to multisite phosphorylation is not evident in the data. The authors should show activities here that are not normalized.

We now provide the absolute activities as Supplementary Fig. 5f.

5) The conclusion on line 198 that “Mutation of Cdk7 T170 to alanine reduced the levels of S164 phosphorylation, whereas the Cdk7 S164A mutation had no effect on pT170 levels” is based on comparison of band intensities in a Western blot. The differences seem subtle, and there is no indication of replicates. A more quantitative assay such as mass spectrometry would better support the conclusion.

We quantified the replicates of the transfection experiments and include them in the new Supplementary Fig. 7c. We apologise for not providing these data in the initial submission.

6) The conclusion on line 200 states that “Mutation of Cdk7 T170 severely reduced activity to levels close to those of binary Cdk7/Cyclin H complexes,” however, data for the binary complex are not shown in Fig. 3b. Perhaps authors are referring to data in Fig. 3d? Similarly, beginning on line 206, a comparison is made to unphosphorylated CDK7, but it is not clear where those data can be found.

We thank the Reviewer for pointing this out. The statement was indeed based on knowledge of activity patterns and is not retrievable from the data presented in Fig. 3b. We apologize for any confusion this may have caused. We changed the configuration of Figure 3 to address this point appropriately. It now shows activity patterns of binary Cdk7/Cyclin H complexes (wt and mutants) and the resulting change upon *in vitro* reconstitution with Mat1. This way, the activities of binary and ternary kinase complexes are directly comparable. The *in vitro* reconstitution matches well with the data obtained from co-purification experiments. A comparison of the activity of the S164A/T170A mutant to unphosphorylated Cdk7 is now provided in the new Supplementary Fig. 4c.

7) A few suggestions for the discussion: The first few paragraphs largely recapitulate structural results and could be condensed or removed. The logic leading to the conclusion in the last sentence is unclear. It may be interesting to discuss the implications of the results for pharmacological inhibition of CDK7.

We agree with the Reviewer and have fully rewritten the first two paragraphs of the Discussion section in the revised manuscript. We deleted the redundant description of the Cyclin H N-terminus and the networks of the two phosphorylation sites. Following the suggestion of the third Reviewer, we now compare our doubly phosphorylated Cdk7 structure with the recently published unphosphorylated Cdk7 structure, which sheds light on the electrostatic surface of the substrate recognition site. The structure reported here reveals an electrostatically neutral surface in proximity to the active site, which is unique to fully active Cdk7/Cyclin H/Mat1 and might afford opportunities as pharmacophoric site for development of new Cdk7-selective small molecules.

Reviewer #3 (Remarks to the Author):

Düster et al. present a CAK crystal structure of a CAK that is completely phosphorylated at the T-loop positions S164 and T170. The 2.15 Å crystal structure was accomplished by diligent and clever usage of nanobodies. The structural data are supplemented with a detailed biochemical analysis concerning different phosphorylation states of the CDK7 T-loop and the effects thereof on RNAP II CTD, SPT5, and CDK2 phosphorylation. The authors can show that CAK with a fully phosphorylated T-loop has the highest activity towards the CTD, whereas CDK2 phosphorylation is almost unaffected. The authors discuss in detail the structural necessities and outcomes of the phosphorylation events in terms of activity.

Although the work is interesting it is important to address several concerns prior to publication.

The biochemical analysis of the complex is very interesting and compelling. However, the authors have not provided a real ground state for their activity measurements. T-loop phosphorylated CDK7/Cyclin H is compared with non-phosphorylated trimeric CAK or fully phosphorylated trimeric CAK that was obtained by mixing lysates but. the analysis of unphosphorylated CDK7/Cyclin H is missing to provide a ground state. The T-loop phosphorylated CDK7/Cyclin H displays already significant activity that is

hardly increased by the presence of MAT1 for the non-phosphorylated complex. Previous data indicate, however, that CDK7 or CDK7/Cyclin H activity is significantly lower than CAK activity (Lolli 2004, Peisert 2020,) in the non-phosphorylated state. It is thus important to provide data for the ground state to obtain a complete picture of the phosphorylation effects on CAK activity.

We now provide additional information on the activities of unphosphorylated Cdk7/Cyclin H. We purified the T170A, S164A single point mutants and the S164A/T170A double mutant in complex with cyclin H and analysed the activity pattern towards CTD in the absence and presence of Mat1 using the *in vitro* reconstitution setup. As expected, the Cdk7 (T170A) mutant was inactive in its binary state, but activity could be recovered by Mat1 incubation. The Cdk7 (S164A/T170A) double mutant was inactive as a binary Cdk/Cyclin complex, but was only mildly stimulated by Mat1. These data are in good agreement with our data obtained from co-purified ternary complexes. The new data are shown in Fig. 3a,b. The former Fig. 3a,b is now moved to Supplementary Fig. 4d,e.

Regarding the activity patterns in the mentioned studies, we would like to highlight that in Lolli et al (2004) there is no analysis of unphosphorylated ternary complexes. The study provides a comparison of pS164/pT170 phosphorylated Cdk7/Cyclin H with partly phosphorylated ternary complexes. This likely arises from the expression protocol. For expression of ternary complexes, *Sf9* cells were infected with individual virus at a low MOI of 0.3. This way, expressions probably contained a mixture of singly, doubly and triply infected cells resulting in a very heterogeneously phosphorylated sample.

The activity data of the ctCDK7 homolog are difficult to interpret as there is no comprehensive biochemical evaluation accompanying the structure. (There is also no evidence that ctCDK7 is a CAK.) The ctCDK7 sample is not phosphorylated in the T-loop and thus likely to be only minimally active in absence of Mat1. The gain in activity upon Mat1 binding reflects the power of Mat1 to induce an active conformation in absence of T-loop phosphorylation. Overall, both mentioned studies are in good agreement with our data.

A serious concern is the description of the CAK structure. The authors mention previously obtained CAK structures but fail to include a detailed comparison in a figure, with special emphasis on T-loop movement and positioning. A detailed comparison would have shown that the T-loop position between all available structures is highly similar. This analysis should be clearly included in the data interpretation especially in the paragraph “dual phosphorylation puts the T-loop in its place” which at least adds a question mark to the subtitle. In fact, the similarity of the structures raises the question why the activity is altered although the T-loop positioning is structurally very similar. The different interactions of the phosphate networks are appreciated. However, in a superposition with the coordinates of human CAK (pdb entry 8orm) of seemingly non phosphorylated CAK the side-chain positions do not seem to differ significantly for the T-loop and the phosphate receiving residues. This should be considered and discussed accordingly. It is also important to mention that the missing first 45 residues of the N-lobe of CDK7 constitute a major part of the ATP binding pocket that is virtually not present in the here presented structure.

We agree with the Reviewer that a thorough comparison to previously obtained CAK structures was missing from the first version of the manuscript and now include a new Figure 7 that shows an overlay of our doubly phosphorylated Cdk7 structure (PDB ID 8pyr) with the monomeric, singly phosphorylated Cdk7 pT170 structure (1ua2) and the recently published high-resolution

cryo-EM structure of the ternary CAK complex with unphosphorylated Cdk7 (8orm). The conformation of the T loop in the latter structure of the kinase is almost identical to that in the doubly phosphorylated Cdk7 structure, such that it might be more appropriate to say that Mat1 puts the T loop in its place (we now change the subtitle to “keeps the T loop in its place”). However, the electrostatic surface potential significantly changes between the unphosphorylated and the doubly phosphorylated kinase from a largely basic one to a balanced charge distribution. This is now described in the first two paragraphs of the Discussion section in the revised version of the manuscript, including the comment that the ATP-binding pocket is incomplete due to the missing first 45 amino acids in the electron density map.

An additional concern is that the biochemical data sometimes seems to be rather inconsistent between different experiments. In Figure 1c the activation of CDK7/Cyclin H by the addition of MAT1 is approx. 4-5 fold, whereas it is 8-10 fold in panels 3d and 3f. The main difference between these experiments seems to be that MAT1 was added later and in 4 fold excess of CDK7 (for the results shown in Figure 3d and 3f). The elevated activity of these complexes could indicate that the initial purification strategy that has also been applied in other experiments in the manuscript, leads to less and unpredictable activity patterns. In Fig. 4a, however, the difference at 15 min seems to be tenfold or maybe even more with the copurification strategy. To overcome this high variability the authors should stick to one method of complex preparation for their biochemistry or clearly state and discuss differences between the approaches. In addition, it seems that the stoichiometry of the complexes presented in panel 3a significantly differs and thus cannot be compared as such. It could be necessary to pursue a different purification strategy? If possible, biological duplicates for every key experiment should be performed, rather than technical replicates especially in light of the issues to obtain an active stoichiometric complex. Overall, the different range of data points for many experiments seems odd. The aim should be to perform most of the experiments with a similar amount of technical and biological replicates.

Regarding the variability of the data, we want to emphasize that different batches of Cdk7 (wild-type and mutants) were tested that all produced similar results. We refer the Reviewer to Figure 3d, right panel, in which we compiled data from several experiments performed under the same conditions. Here we show that the factor of stimulation for wild-type Cdk7 by Mat1 within a single experiment (performed in duplicate) ranges from 5- to 14-fold with a mean of 8.8. The factor by which the samples mentioned in Fig. 1c differed is 5.7, which is therefore within the expected range of stimulation, albeit towards the lower end. These data were collected on different days and with at least two different kinase batches each.

We revised Fig. 3 for more consistency to address the Reviewer’s concern regarding different complex preparation strategies for activity determination. Fig. 3 now exclusively shows activity data from *in-vitro* reconstitutions of binary Cdk7/Cyclin H complexes and Mat1. This way we can dissect the influence of T-loop phosphorylation and Mat1 binding on kinase activity individually. Another advantage of this strategy is that activity profiles of binary and matching ternary samples are less influenced by technical issues (e.g., freezing or storing of samples) since the underlying kinase preparation is the same for both samples.

In Fig. 4 we switched to co-purified samples. We agree that the strategy to obtain unphosphorylated samples is not airtight and thus there appears to be some heterogeneity in Cdk7 phosphorylation state (which is shown in Supplementary Fig. 1). However, co-expression is the only way to generate predominantly unphosphorylated Cdk7 preparations without mutating the T-loop. This is of particular importance, as we show in Fig. 3b and Supplementary

Fig. S4c that Cdk7 S164A/T170A mutant complexes display artificially low activity that does not reflect the unphosphorylated WT state.

The authors may consider to show the concentration dependent activity of the analyzed protein complexes since the enzyme to substrate ratio used here is 1:100 which is rather high on the substrate side. A complementary method like a coupled ATPase assay would provide more kinetic data than the 1 point measurements used here and also be helpful to assess the quality of the obtained data.

We thank the Reviewer for this point, which was also raised by Reviewer #2. We now provide data of ATP titration and CTD titration with different Cdk7 preparations in the new Supplementary Fig. 5a-c.

Finally, the discussion section should address differences and similarities to previous structures especially the T-loop position and the phosphate networks.

We agree with the Reviewer and included a new paragraph on the differences and similarities to previous Cdk7 structures (PDB IDs 1ua2, 8orm) in the revised version of the manuscript. This suggestion led to a new Figure 7, which specifically addresses changes in the electrostatic surface potential of unphosphorylated Cdk7 compared to doubly phosphorylated Cdk7. Following also the second Reviewer's suggestion some redundant descriptions of the pS164/pT170 network have made way for this new section.

Minor issues:

In panel 6a a more annotation of the different bands would help the reader to understand the meaning of the figure.

We revised Fig. 6 accordingly and added additional information in the new Supplementary Fig. 7 for more clarity.

In panel 2b all side chains of amino acids are visible except the one of F162. For consistency this should also be shown.

We thank for this attentive comment and now display also the side chain of F162 in the revised version of the manuscript.

When the effect of pre-existing phosphorylation on the kinase activity was investigated (panel 4e and 5c in the supplements), the variant pS5-mid was omitted without reason. Furthermore, background information of why Ser7 was exchanged to a lysine would be helpful.

We used the peptides to test if we can get insight into the directionality of the phosphorylation reaction and potential alterations upon Cdk7 T loop phosphorylation. However, the peptides were purchased as a custom synthesis for a different project. At that time, pS5-mid was not

ordered. Since there is no difference between the Cdk7 preparations we decided not to pursue this further.

Ser7 to Lys is the most common alteration of the human CTD in the distal part. We now mention this in the text. We thank the Reviewer for reminding us that this cannot be regarded as common knowledge.

Please, correct the typo in the caption of panel 2f in the supplements since the crystal structure was obtained using VHH(RD7-04) instead of VHH(RD7-01), as correctly shown in the figure itself.

Thanks for this very attentive comment, done!

We thank all Reviewers for their considerate comments and kind assessment of our study.

REVIEWERS' COMMENTS

Reviewer #1 (Remarks to the Author):

I thank the authors for their explanations, discussion, and additional experiments to address my previous suggestions. I do not have any further comments. I find this manuscript meets the standards and I support its publication in Nature Communications.

Reviewer #2 (Remarks to the Author):

The authors have carefully considered reviewer concerns and included additional data and analysis that improve the manuscript. The potential impact of the study remains somewhat limited in that the revisions still do not provide meaningful mechanistic insights explaining how T loop phosphorylation contributes to Mat1 stimulation of kinase activity or CTD phosphorylation. Nevertheless, the conclusions are all now well supported by the data and will be of interest, especially in that the study clarifies the significance of S164 phosphorylation.

Reviewer #3 (Remarks to the Author):

My comments have been addressed and I now recommend to accept the manuscript for publication.